# KAN-GLNet: An enhanced PointNet++ model for canola silique segmentation and counting

**Jiajun Liu**[1], **Bei Zhou** [1,2]*, **Jie Liu**[3], **Xike Zhang**[1], **Jiangshu Wei**[1,2], **Yao Zhang**[1], **Junjie Wu**[1], **Changping Wu**[3], **Di Hu**[1]

**1** College of Information Engineering, Sichuan Agricultural University, Yaan, China, **2** Sichuan Key Laboratory of Agricultural Information Engineering, Sichuan Agricultural University, Yaan, China, **3** College of Agronomy, Sichuan Agricultural University, Chengdu, China

* 12801@sicau.edu.cn

## Abstract

Accurate analysis of plant phenotypic traits is crucial for crop breeding and precision agriculture. This study proposes a lightweight semantic segmentation model named KAN-GLNet (Kolmogorov–Arnold Network with Global–Local Feature Modulation), based on an enhanced PointNet++ architecture and integrated with an optimized Density-Based Spatial Clustering of Applications with Noise (DBSCAN) algorithm, to achieve high-precision segmentation and automatic counting of canola siliques. A multi-view point cloud acquisition platform was built, and high-fidelity canola point clouds were reconstructed using Neural Radiance Fields (NeRF) technology. The proposed model includes three key modules: Reverse Bottleneck Kolmogorov–Arnold Network Convolution, a Global–Local Feature Modulation (GLFN) block, and a contrastive learning-based normalization module called ContraNorm. KAN-GLNet contains only 5.72M parameters and achieves 94.50% mIoU, 96.72% mAcc, and 97.77% OAcc in semantic segmentation tasks, outperforming all baseline models. In addition, the DBSCAN workflow was optimized, achieving a counting accuracy of 97.45% in the instance segmentation task. This method achieves an excellent balance between segmentation accuracy and model complexity, providing an efficient solution for high-throughput plant phenotyping. The code and dataset have been made publicly available at: https://anonymous.4open.science/r/KAN-GLNet-6432/.

## Introduction

Plant phenotype, determined by the interaction between plants and their environment, serves as one of the key information sources for studying and managing plant growth and development [1]. As a globally important oil crop, rapeseed is widely cultivated due to its economic and ecological value [2]. Yield is primarily determined by silique number, seeds per silique, and thousand-seed weight [3], with silique number being the most critical factor. However, during the Silique maturation stage, the dense and overlapping distribution of siliques makes accurate identification extremely

**Data availability statement:** Our curated code and dataset can be accessed at the following link: https://anonymous.4open.science/r/KAN-GLNet-6432/.

**Funding:** This project is supported by National Natural Science Foundation of China, grant number 32301762.

challenging. While deep learning models can handle this complexity, their high parameter count incurs substantial storage and computational costs, limiting high-throughput analysis. Traditional manual counting methods are inefficient and highly subjective. Therefore, achieving high-precision recognition with low-parameter models in complex plant scenarios is crucial for overcoming computational bottlenecks and advancing precision agriculture.

Traditional point cloud generation technologies each have distinct characteristics: LiDAR achieves millimeter-level precision but involves high hardware costs; depth cameras enable real-time acquisition but are susceptible to lighting interference; multi-view stereo (MVS) has low equipment costs but requires hours of computational processing. In contrast, Neural Radiance Fields (NeRF, [4]) represent a technological breakthrough through implicit neural representation—experiments demonstrate its reconstruction accuracy matches MVS levels [5]. The improved Nerfacto model [6] not only significantly reduces training time but also achieves superior geometric fidelity. Building on this, our study establishes the first fully annotated rapeseed NeRF dataset, providing an innovative high-precision, low-cost solution for 3D phenotyping of complex plant organs.

Accurate organ structure segmentation forms the foundation for quantifying morphological phenotypic parameters from 3D point cloud data. Traditional semantic segmentation methods employ geometric topological models, handcrafted features, and prior knowledge about plant objects to describe and distinguish different plant organs. For instance, Wu Sheng et al. [7] extracted maize skeletons across growth stages using Laplacian contraction combined with adaptive sampling, calculating plant height, leaf length, and inclination angles. Paloma Sodhi et al. [8] achieved precise sheath-stem segmentation and phenotyping in sorghum through multi-view 3D reconstruction and SVM classification with local-global feature fusion. Li Dawei et al. [9] proposed a region-growing segmentation method involving iterative PCA-based spatial feature calculation, supervoxel over-segmentation, and region growth for greenhouse plant leaf segmentation. However, these methods heavily rely on prior morphological knowledge, require tedious parameter tuning, are noise-sensitive, and have limited capability for analyzing complex structures and traits [10–12].

To address these challenges, deep learning-based point cloud segmentation methods have emerged, automatically learning features through data-driven approaches. Early research primarily adopted voxelization to convert point clouds into structured data for feature extraction [13]. For example, Das Choudhury et al. [14] segmented maize stems and leaves using multi-view visual hull algorithms with voxel overlap verification and Euclidean clustering. Saeed et al. [15] implemented 3D segmentation of cotton stems, branches, and bolls via Point-Voxel Convolutional Neural Networks (PVCNN). However, these methods demand substantial computational resources and may incur information loss during segmentation. To overcome point cloud processing bottlenecks, subsequent research focused on direct point cloud processing architectures, mainly developing along two directions: point-based methods and Transformer-based approaches. The pioneering PointNet series [16,17] laid the groundwork for point cloud deep learning, providing crucial technical support for plant

3D phenotyping. For instance, Ao et al. [18] applied PointNet for maize stem-leaf separation using local point density. Guo et al. [19] integrated PointNet++ with ASAP attention modules, achieving 86% mIoU in cabbage segmentation. Addressing PointNet++ optimization needs, PointNeXt achieved performance breakthroughs through training strategy updates and model scaling, with Dong et al. [20] reporting 89.21% mIoU (sugarcane), 89.19% mIoU (maize), and 83.05% mIoU (tomato) stem-leaf segmentation at 6.08M parameters. While such point-based models excel in parameter efficiency, they often face accuracy limitations due to insufficient feature extraction. PCT [21] pioneered Transformer architecture for 3D point cloud processing, marking its first application in point cloud segmentation. For example, Ma et al. [22] proposed PSTNet with cascaded self-attention (PSA) and local feature aggregation (NPA), achieving 92.20% IoU for eggplant point clouds. Yang et al. [23] developed PACANet, a Transformer-based pairwise attention center axis aggregation network, achieving 92.46% mean accuracy for maize populations at 46.2M parameters. These studies demonstrate that while Transformer architectures deliver breakthrough performance, they often suffer from parameter explosion, severely limiting deployment in resource-constrained scenarios.

To address the contradiction between parameter quantity and segmentation accuracy in plant point cloud segmentation, this study proposes KAN-GLNet, a Kolmogorov-Arnold segmentation network equipped with global-local feature modulation. The model is designed to achieve high-precision segmentation under low-parameter constraints. The key innovations include:

(i) Constructing the first NeRF-derived rapeseed silique point cloud dataset (50 samples), expanding sample scale through data augmentation strategies.

(ii) Proposing reverse bottleneck KAN convolution (enhanced Kolmogorov-Arnold convolution) to replace conventional convolutions for strengthening feature extraction capability.

(iii) Designing GLFN multi-scale feature modulation blocks to optimize spatial representation of complex silique structures by fusing global-local attention mechanisms.

(iv) Introducing the ContraNorm contrastive normalization module to enhance point cloud feature stability and segmentation consistency.

(v) Proposing an optimized DBSCAN workflow algorithm to achieve automated silique instance counting.

## Materials and methods

### Overview

The overall process of canola silique segmentation and counting is shown in Fig 1, which consists of five stages: data acquisition, 3D point cloud reconstruction, data preprocessing and augmentation, canola silique segmentation, and silique counting.

### Data collection

The canola plants used in this study were cultivated in May 2024 at the No. 91 experimental field of Sichuan Agricultural University, Ya'an City, Sichuan Province, China, located at 30°N 103°E. To achieve high-throughput data acquisition, we employed a multi-angle photography approach, aiming to comprehensively capture the structural features of the canola plants [24]. As shown in Fig 2, the custom imaging platform consisted of four core components: a smartphone, two sets of supplemental lighting equipment, an electric turntable, and a black backdrop. Given the inherent limitations of smartphone image sensor sensitivity, the lighting system served a critical function. The turntable precisely controlled shooting angles while the backdrop effectively eliminated background interference.

Data collection employed an Apple iPhone 15 Pro Max smartphone featuring a 48-megapixel main camera with an f/1.78 aperture and 24mm focal length. The study involved standardized acquisition of 50 canola plants, each positioned centrally on a turntable rotating at a constant angular velocity of 6 degrees per second. The camera maintained a fixed

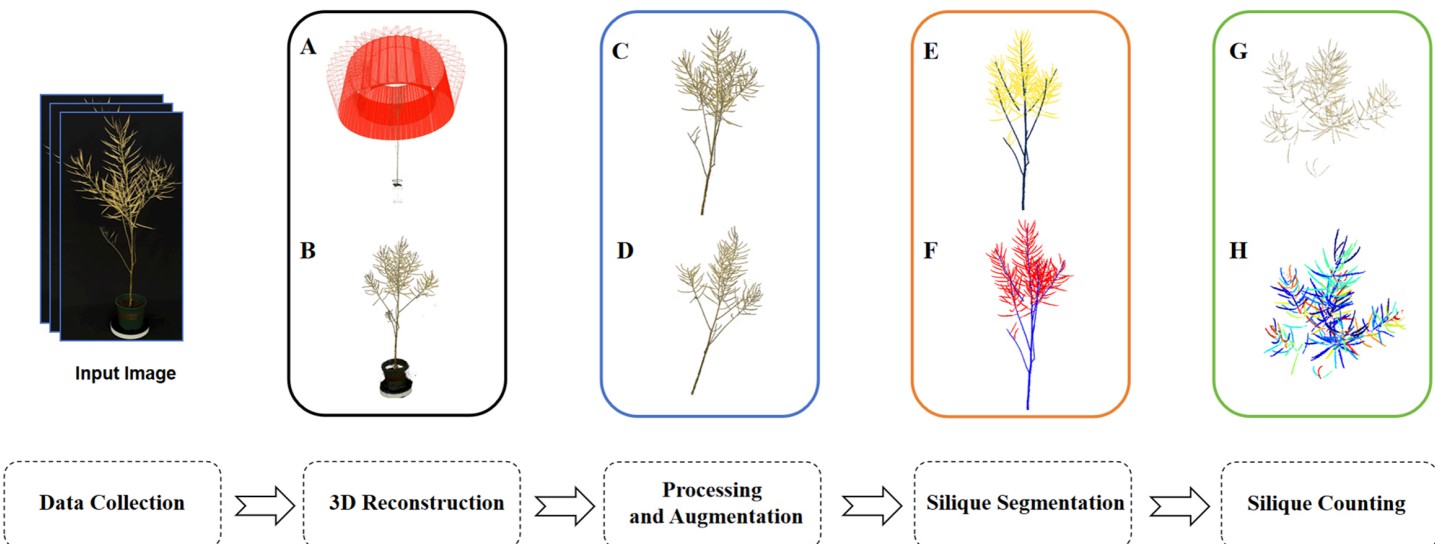

**Fig 1**. **Overview of canola silique segmentation and counting.** (A) Sparse reconstruction. (B) Dense reconstruction. (C) Point cloud preprocessing. (D) Point cloud augmentation. (E) Segmentation model prediction result. (F) Segmented Silique Point Cloud. (G) Silique instance clustering result.

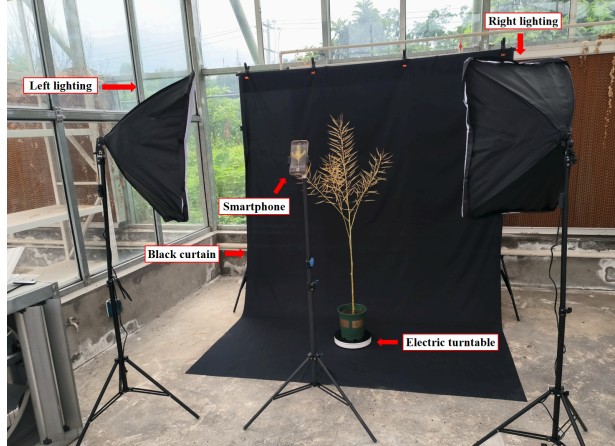

**Fig 2**. **Imaging platform.**

horizontal distance of 50 centimeters from the subject, systematically capturing video recordings from two distinct perspectives: a horizontal view and a 60-degree downward-tilted view. Each perspective was recorded for one minute, yielding two minutes of total footage per plant. The description of acquisition parameters was inspired by [25], particularly in terms of presenting camera specifications and shooting settings. Complete parameter details are provided in Table 1.

For image processing, FFmpeg software [26] extracted keyframes at one-second intervals, generating 120 frames per plant at a resolution of 3840 by 2160 pixels in JPG format, suitable for subsequent 3D reconstruction and analysis. The turntable's continuous rotation protocol ensured optimal imaging stability while effectively preventing motion blur artifacts.

**Table 1**. Rapeseed video acquisition and image processing data protocol.

| Parameter | Details |
|---|---|
| Camera Type | High-resolution digital camera |
| Brand & Model | Apple iPhone 15 Pro Max |
| Camera Specifications | 48MP main + 12MP ultra-wide + 12MP 2x/5x telephoto |
| Aperture | f/1.78 (main) |
| Focal Length | 24mm (main) |
| Environment | Windless indoor with artificial lighting |
| View Angles | Horizontal + 60° downward tilt |
| Recording Duration | 1 min per angle (2 min total) |
| Shooting Distance | 50cm (lens-to-object horizontal) |
| Turntable Speed | 6°/s |
| Video Format | 4K/60fps (3840×2160 pixels) |
| Storage Format | HEVC/H.264 encoding |
| ISO Sensitivity | 25-6400 (auto) |
| Video Frame Extraction Method | FFmpeg keyframe sampling (1 fps) |
| Video Frame Resolution | 3840×2160 pixels |
| Output Image Format | JPG |

## Point cloud generation and processing

NeRF is a technique that utilizes neural networks to represent scenes in three dimensions by learning volumetric representations from multi-view 2D images, thereby producing high-quality 3D reconstructions. The reconstruction process includes image acquisition, ray sampling, volume rendering, loss calculation and network optimization, volume density extraction, and point cloud generation, ultimately culminating in the completion of the 3D reconstruction.

NeRFStudio [27] is a framework designed for the creation, training, and deployment of NeRF models, aimed at simplifying the application process of NeRF technology, thus enabling researchers to utilize this technique more easily. Among these models, Nerfacto combines the advantages of traditional NeRF with the latest technological advancements, enhancing the speed, quality, and flexibility of 3D reconstruction. In this study, the Nerfacto model was employed to generate the point cloud data for canola. This method boasts a high reconstruction efficiency, with each object requiring approximately 20 minutes for reconstruction.

In the reconstruction process, images of the canola from multiple angles were loaded and synchronized. COLMAP [28] was used for feature matching and pose estimation to generate sparse point clouds and camera poses, which were then input into the Nerfacto model to create high-quality dense point clouds. These point clouds were saved in a P × 6 array, where P represents the total number of points, with columns for spatial coordinates (X, Y, Z) and corresponding RGB values, stored in PLY format. Fig 3 displays multiple sets of canola point cloud data generated by the Nerfacto model.

Nerfacto-generated point cloud models are highly dense, with each canola plant's 3D model consisting of 200,000 to 300,000 points. These models initially included not only the canola plants but also trays, turntables, and significant noise points. As the main goal was to segment the canola siliques, the tray and turntable components were manually removed using CloudCompare [29], and noise was removed using radius-based and statistical methods due to the numerous outliers.

After denoising, a uniform downsampling method was applied to preserve the point cloud's distribution characteristics and essential information [30]. The point cloud for each canola plant was downsampled to 20,000–30,000 points to reduce data volume and improve computational efficiency [31].

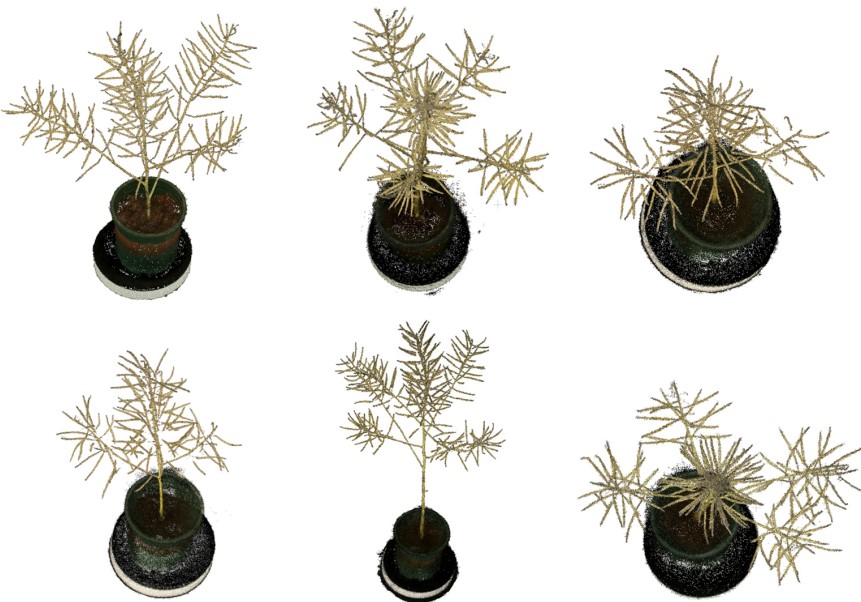

**Fig 3**. **Reconstructed canola point cloud samples.**

## Methodology

### Network overview

PointNet++, as a mainstream framework for 3D point cloud analysis, extracts local and global features through its Set Abstraction (SA) and Feature Propagation (FP) mechanisms. However, it has limitations in modeling long-range dependencies and insufficient global feature representation. To address these issues, this study proposes the KAN-GLNet model (Fig 4), which reconstructs the Min-PointNet module in the SA layer into KGL-PointNet, achieving improvements in three aspects: First, we propose reverse bottleneck KAN convolutions to replace some of the original convolutional layers, which achieves more efficient geometric feature extraction with fewer parameters compared to standard KAN convolutions [32]; second, a GLFN feature modulation block is designed, combining Global and Local Spatial Attention (GLS Attention) with a Partial Convolution Network (PCFN) to jointly optimize local details and global contextual representation; finally, the ContraNorm contrastive normalization module [33] is introduced, leveraging contrastive learning to constrain feature distributions, suppress noise, and mitigate dimensional collapse and over-smoothing during training. The input point cloud batch has a feature dimension of D, with K neighboring points selected for each sampled point and N total sampled points. The overall architecture of KGL-PointNet and its core components are shown in Fig 5.

### Reverse bottleneck KAN convolutions

The recently proposed Kolmogorov-Arnold Networks (KANs) have emerged as a promising alternative to MLPs [34], prompting us to examine them closely. KANs are a type of neural network architecture inspired by the pioneering theories of Andrey Kolmogorov and Vladimir Arnold. Unlike traditional MLPs, KANs replace linear weights with learnable spline functions. The advantage of this approach is that it not only reduces the number of required parameters but also improves the generalization ability of the network to some extent.

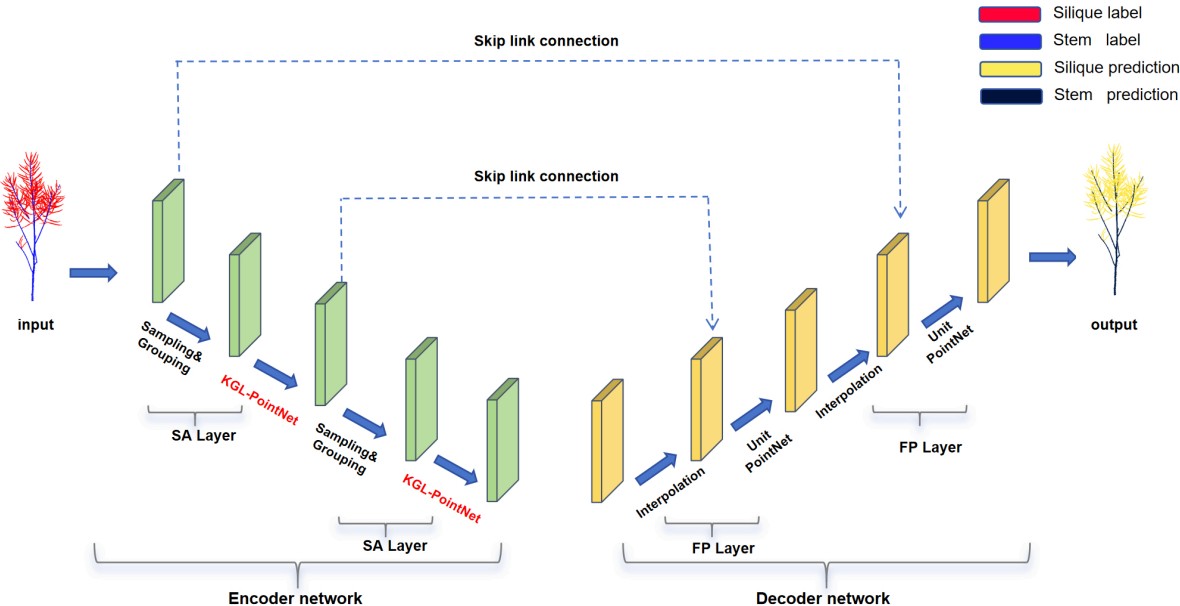

**Fig 4**. **KAN-GLNet network model.**

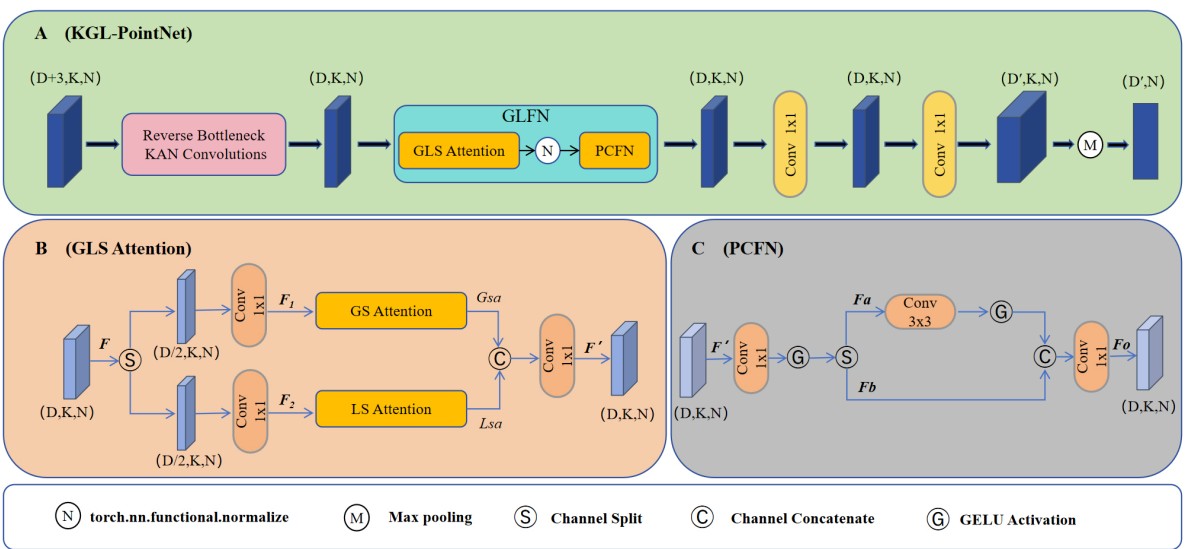

**Fig 5**. **The overall architecture of KGL-PointNet and its core components.** (A) The complete network structure of KGL-PointNet. (B) The multi-scale feature enhancement module, GLS Attention. (C) The feedforward network based on partial convolution, PCFN.

Bodner et al. [32] proposed Kolmogorov-Arnold Convolutions, which are defined as follows. Let the input image be $y$, where $y \in \mathbb{R}^{c \times h \times w}$, $c$ denotes the number of input channels, and $h$ and $w$ are the height and width of the image, respectively. The Kolmogorov-Arnold Convolution with a kernel size of $k$ is defined in Eq (1):

$$x_{ij} = \sum_{d=1}^{c} \sum_{a=0}^{k-1} \sum_{b=0}^{k-1} \varphi_{a,b,d}(y_{d,i+a,j+b}); \quad i = \overline{1, h-k+1}, \quad j = \overline{1, w-k+1} \tag{1}$$

The function $\varphi$ is a univariate nonlinear learnable function. Its specific form is given in Eqs (2) and (3), as defined in the original KANs paper [34]:

$$\phi(x) = w_b\, b(x) + w_s\, \text{spline}(x) \tag{2}$$

$$b(x) = \text{SiLU}(x) = \frac{x}{1 + e^{-x}} \tag{3}$$

Here, $w_b$ and $w_s$ are trainable weights used to control the overall scale of the function. The term spline($x$) represents a spline function, which by default is a linear combination of B-spline basis functions.

In [34], B-splines are employed to approximate smooth functions. However, during training, variables may fall outside the predefined domain, which requires rescaling the spline grid. Although this method is theoretically sound, computing B-spline basis functions and rescaling the grid can introduce computational inefficiencies in KAN-based networks. To address this issue, Li [35] proposed FastKAN, which uses Gaussian radial basis functions (RBFs) to approximate the B-spline basis and thus accelerates model training. The Gaussian RBF is defined in Eq (4):

$$\text{spline}(x) = \exp\left(-\frac{r^2}{2\sigma^2}\right) \tag{4}$$

where $r$ denotes the radial distance and $\sigma$ is the standard deviation controlling the width of the Gaussian function.

However, although FastKAN accelerates model training, it does not solve the issue of KANs' reliance on a large number of parameters during training, which still results in high training costs and increases the likelihood of overfitting. The problem of large convolutional parameter counts in KANs mainly arises in the *spline* part. Simply replacing the basis functions introduces a significant number of parameters into the model.

To overcome the limitations of existing methods, we propose Reverse Bottleneck KAN, which uses Gaussian kernel RBFs to approximate the B-spline basis. Unlike FastKAN, we remove the direct large-kernel convolution on the spline and adopt a reverse bottleneck structure. Specifically, this method first expands the dimensionality of the input data using a convolution with a kernel size of 1, then applies the *spline*, and finally reduces the dimensionality with another convolution of kernel size 1. This design, similar to a single-layer encoding-decoding process, improves feature representation while reducing the number of parameters.The network architectures of Kolmogorov-Arnold, FastKAN, and Reverse Bottleneck KAN Convolutions are shown in Fig 6.

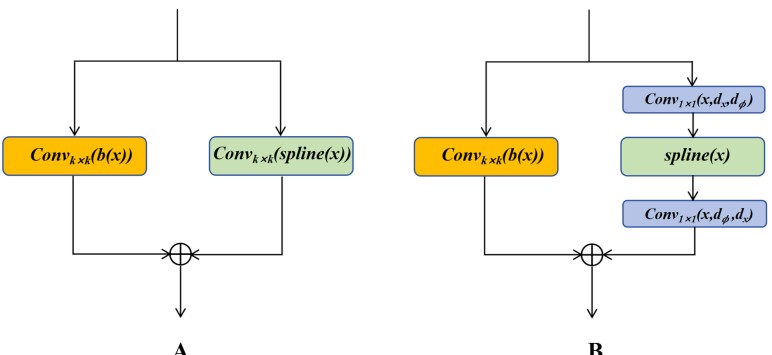

**Fig 6**. **Network architecture diagrams of Kolmogorov-Arnold, FastKAN, and reverse bottleneck KAN convolutions.** (A) Kolmogorov-Arnold and FastKAN Convolutions. (B) Reverse Bottleneck KAN Convolutions.

## GLFN feature modulation block

Although Reverse Bottleneck KAN Convolutions can extract initial features and generate basic representations from raw input, these features often struggle to adequately and finely represent key information in the point cloud, especially in terms of detail and precision. To further enhance the network's ability to extract key features, we draw on the attention mechanism from [36] and the feature fusion method from [37], applying them innovatively to the 3D point cloud domain. To achieve this, we design the GLFN feature modulation module, designed to enhance feature extraction after the Reverse Bottleneck KAN Convolutions.

The GLFN module consists of the GLS Attention module and PCFN. The GLS Attention module enhances multi-scale feature representation, while the PCFN refines and denoises features using partial convolution, integrating local and global information. The module takes the output from Reverse Bottleneck KAN Convolutions as input and enhances the network's ability to capture local geometry and global context through multi-level feature modulation. This design improves the expression of local details and global features, significantly boosting the overall performance of the network.

**GLS attention.** The attention mechanism enhances relevant information while suppressing irrelevant details. The proposed GLS Attention module consists of two components: Global Spatial Attention (GS Attention) and Local Spatial Attention (LS Attention), which collaboratively enhance features at different spatial scales.

Given a feature tensor $F$ with $D$ channels, we first split it into two parts, $F_1$ and $F_2$, as shown in Eq (5). These two tensors are then passed through GS and LS Attention modules, respectively. Their outputs are concatenated and fused via a $1 \times 1$ convolution, as defined in Eq (6):

$$F_1, F_2 = \text{Split}(F) \tag{5}$$

$$F' = C_{1 \times 1}\left(\text{Concat}\left(Gsa(F_1), Lsa(F_2)\right)\right) \tag{6}$$

Here, $Gsa(\cdot)$ and $Lsa(\cdot)$ denote the Global and Local Spatial Attention modules, respectively, and $F'$ is the final output feature after fusion via $1 \times 1$ convolution. The detailed structures of the GS Attention and LS Attention modules are illustrated in Fig 7.

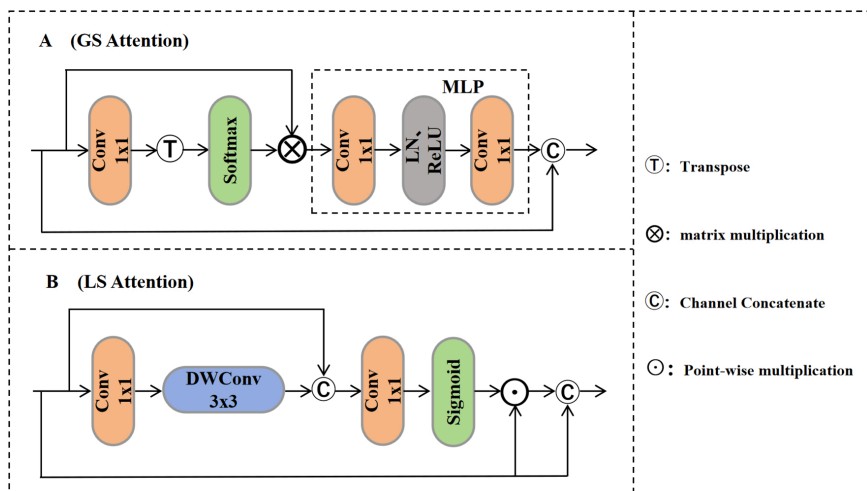

**Fig 7**. **Network architecture diagrams of the GS attention module and the LS attention module.** (A) GS Attention module structure. (B) LS Attention module structure.

(1) **GS Attention Module**: The GS Attention module captures long-range dependencies between points in the spatial dimension of point clouds, thereby complementing the local spatial attention. Prior studies [38,39] have shown that such long-range interactions significantly enhance feature representation. The global spatial attention is computed using the feature tensor $F_1$, and the corresponding operations are defined in Eqs (7) and (8):

$$\text{Att}_G(F_1) = \text{Softmax}\left(\text{Transpose}\left(C_{1\times1}(F_1)\right)\right) \tag{7}$$

$$Gsa(F_1) = \text{MLP}\left(\text{Att}_G(F_1) \otimes F_1\right) + F_1 \tag{8}$$

Here, $\text{Att}_G(\cdot)$ denotes the global spatial attention operator, $C_{1\times1}$ is the $1 \times 1$ convolution, $\otimes$ indicates matrix multiplication, Transpose represents transposition, and $\text{MLP}(\cdot)$ includes two fully connected layers, a ReLU activation, and a normalization layer.

(2) **LS Attention Module**: The LS Attention module enhances local region features in point clouds, especially beneficial for highlighting small objects. We compute the local spatial attention and apply it to the tensor $F_2$, as shown in Eqs (9) and (10):

$$\text{Att}_L(F_2) = \text{Sigmoid}\left(C_{1\times1}(Fc(F_2)) + F_2\right) \tag{9}$$

$$Lsa(F_2) = \text{Att}_L(F_2) \odot F_2 + F_2 \tag{10}$$

In this context, $Fc(\cdot)$ is a cascaded block composed of three $1 \times 1$ convolution layers and one $3 \times 3$ depthwise convolution. The operator $\odot$ denotes element-wise multiplication, and $\text{Att}_L(\cdot)$ represents local spatial attention. This design effectively captures fine-grained local features with minimal parameters.

**PCFN module.** Although the GLS Attention module enhances multi-scale features via global and local spatial attention, it may fail to fully integrate the relationships between them, potentially leading to weak representation and sensitivity to noise. To address this, we introduce the PCFN module—an efficient part-convolutional feedforward network aimed at refining features, reducing noise, and balancing local-global integration.

As illustrated in Fig 7(C), the PCFN module begins by receiving the fused feature $F'$, which is normalized and passed through a $1 \times 1$ convolution followed by a GELU activation. The resulting hidden representation is then split into two feature blocks $\{F_a, F_b\}$, as defined in Eq (11). One of them, $F_a$, undergoes a $3 \times 3$ convolution and GELU activation to encode local context. Finally, the outputs are concatenated and projected back to the original dimension through another $1 \times 1$ convolution, as described in Eq (12):

$$\{Fa, Fb\} = S\left(G\left(C_{1\times1}\left(\|F'\|_2\right)\right)\right) \tag{11}$$

$$Fo = C_{1\times1}(Concatenate[G(C_{3\times3}(F_a)), F_b]) \tag{12}$$

## Normalization

KAN-GLNet is influenced by the PointNet++ baseline model and applies Batch Normalization (BN) after the SA and FP layers to improve model stability, accelerate training, and prevent gradient vanishing or explosion. However, BN may lead to over-smoothing, causing the features to become similar and failing to retain the geometric differences between different point clouds, thereby reducing classification and segmentation accuracy. Moreover, since BN normalizes features only within each batch, it may lead to dimensional collapse, further limiting the network's expressive power.

Inspired by [33], we adopt the contrastive learning-based normalization technique, ContraNorm, for 3D point cloud features. This method effectively mitigates issues such as dimensional collapse and over-smoothing, thereby enhancing the

network's capability to model complex geometric structures. The forward operation of ContraNorm is defined in Eq (13):

$$H_t = \text{LN}\left(H_b - \frac{s}{\tau} \cdot \text{Softmax}(H_b H_b^T) H_b\right) \tag{13}$$

Here, $H_b$ denotes the input feature matrix, and $H_b^T$ is its transpose. The scalar $s$ represents the stride, and $\tau$ is a temperature parameter that controls the strength of the contrastive term. The Softmax function normalizes the similarity matrix $H_b H_b^T$, ensuring the similarity distribution is stable and bounded. The final output $H_t$ is passed through a Layer Normalization (LN) to maintain numerical stability and scale consistency.

## Canola silique recognition based on the DBSCAN algorithm

In plant phenotyping studies, applying clustering algorithms to the silique point clouds extracted after semantic segmentation is a common approach for achieving organ-level 3D point cloud recognition [40]. Elnashef et al. [41] employed the DBSCAN algorithm and utilized local point cloud density features to accomplish high-precision instance segmentation of the stem-leaf structures in dicotyledonous plants. Guo et al. [19], based on semantic segmentation of seedling-stage cabbage point clouds, combined DBSCAN with color filtering and edge filtering to successfully extract a variety of phenotypic traits.

However, directly applying the above clustering methods to canola silique point clouds still faces significant challenges. First, canola siliques often appear densely clustered, interlaced, and heavily overlapped, making it difficult to distinguish between adjacent clusters. Second, the semantic segmentation stage inevitably involves some misclassification between siliques and stems. These incorrect labels introduce additional noise into the clustering input, and outlier points further disrupt cluster connectivity.

Given that canola silique point clouds often contain outliers caused by segmentation errors, occlusions, or background interference, directly performing clustering tends to result in blurred or incorrect cluster boundaries. Therefore, to improve the stability and robustness of the clustering stage, we introduced Statistical Outlier Removal (SOR) filtering after semantic segmentation to clean the silique point cloud before applying DBSCAN clustering. This process effectively improves the quality of the point cloud prior to clustering and reduces the risk of outlier noise being misidentified as siliques.

## Evaluation metrics

To evaluate the performance of our method in 3D semantic segmentation of canola siliques during the silique maturation stage, we adopted widely used metrics, including Overall Accuracy (OAcc), Class Accuracy (Acc), Mean Class Accuracy (mAcc), and Mean Intersection over Union (mIoU). These metrics assess the correctness and overlap between predicted and ground-truth point labels.

In addition, to evaluate the clustering results of individual siliques, we employed three metrics: Mean Absolute Error (MAE), Root Mean Square Error (RMSE), and Counting Accuracy (CA), which measure the numerical deviation between predicted and actual instance counts.

The definitions and formulations of all these evaluation metrics are summarized in Table 2. Here, $n_{ij}$ denotes the number of points of ground-truth class $i$ predicted as class $j$, and $C$ is the total number of semantic classes.

## Experiments

The difficulty of acquiring 3D point cloud data is significantly higher than that of 2D images, which is mainly reflected in two aspects: first, it requires specialized sensors and specific acquisition methods, and second, the complex structure of plants necessitates extensive manual annotation efforts [42]. Data augmentation techniques can effectively alleviate the issue of insufficient training data, providing ample data support for model optimization. The PointNext study

**Table 2. Definitions of evaluation metrics used for semantic segmentation and clustering.**

| Metric | Definition |
|---|---|
| Overall Accuracy (OAcc) | $OAcc = \dfrac{\sum_{i=1}^{C} n_{ii}}{\sum_{i=1}^{C} \sum_{j=1}^{C} n_{ij}}$ |
| Class Accuracy ($Acc_i$) | $Acc_i = \dfrac{n_{ii}}{\sum_{j=1}^{C} n_{ij}}, \quad i = 1, 2, \ldots, C$ |
| Mean Class Accuracy (mAcc) | $mAcc = \dfrac{1}{C} \sum_{i=1}^{C} Acc_i$ |
| IoU per class ($IoU_i$) | $IoU_i = \dfrac{n_{ii}}{\sum_{j=1}^{C} n_{ij} + \sum_{j=1}^{C} n_{ji} - n_{ii}}, \quad i = 1, 2, \ldots, C$ |
| Mean IoU (mIoU) | $mIoU = \dfrac{1}{C} \sum_{i=1}^{C} IoU_i$ |
| Mean Absolute Error (MAE) | $MAE = \dfrac{1}{n} \sum_{i=1}^{n} |y_i - \hat{y}_i|$ |
| Root Mean Square Error (RMSE) | $RMSE = \sqrt{\dfrac{1}{n} \sum_{i=1}^{n} (y_i - \hat{y}_i)^2}$ |
| Counting Accuracy (CA) | $CA = \dfrac{1}{n} \sum_{i=1}^{n} \left( \dfrac{\hat{y}_i}{y_i} \right) \times 100\%$ |

Note: $n_{ij}$ denotes the number of points of ground-truth class $i$ predicted as class $j$, $C$ is the number of classes, $y_i$ is the ground-truth count, and $\hat{y}_i$ is the predicted count for sample $i$.

confirmed the effectiveness of basic augmentation strategies such as rotation and translation in improving model performance. Yao et al. [43] employed random point deletion, noise addition, and scaling for point cloud data of tomato plants across three key growth stages (seedling, flowering, and fruiting). Xie et al. [44] successfully expanded the dataset size to 10 times its original scale by applying operations such as point cloud cropping and jittering to public datasets like Plant3D [45].

Based on previous research experience and computational efficiency considerations, this study adopts a rigorous data processing workflow: first, the NeRF canola dataset is divided into training, validation, and test sets at a ratio of 7:1:2, followed by 10-fold data augmentation for each subset. Splitting the dataset before performing augmentation is to mitigate potential data leakage risks. The specific augmentation strategy includes four dimensions: randomly discarding 0–30% of point cloud data, random rotation along the Z-axis between −180° and 180°, random scaling at a ratio of 0.9–1.1, and a 20% probability of flipping along the Y-axis.

The experiments in this study were conducted in an Ubuntu 20.04 operating system environment, with an NVIDIA RTX 3090 GPU as the hardware configuration. Detailed hardware and software specifications are shown in Table 3. During model training, the number of training epochs was set to 250 to ensure sufficient convergence and performance optimization. The batch size was set to 16, with an initial learning rate of 0.0001, which was dynamically adjusted using a cosine annealing schedule, with a minimum learning rate of 0.00001. The optimizer used was AdamW, with a weight decay coefficient of 0.01 to effectively alleviate overfitting. To improve training efficiency, each batch processed 4,096 sampled points.

**Table 3. Hardware and software specifications.**

| Category | Configuration |
|---|---|
| CPU | Intel(R) Xeon(R) Platinum 8362 |
| GPU | NVIDIA RTX 3090 |
| Memory | 64 GB |
| Deep Learning Framework | PyTorch 2.0.0 |
| Operating System | Ubuntu 20.04 |
| CUDA Version | 11.8 |

## Semantic segmentation results

KAN-GLNet exhibited a clear convergence trend in both training and validation losses. As shown in Fig 8, the loss dropped rapidly during the first 100 epochs, indicating a fast parameter optimization process. Subsequently, the loss curves gradually stabilized, with the validation loss reaching a low plateau while the training loss continued to decline slowly, ultimately achieving full convergence at epoch 250. Although there was a slight gap between the training and validation loss and accuracy curves, the difference was minimal, primarily due to the limited number of original training samples (only 45). Overall, even under the constraint of limited sample size, KAN-GLNet demonstrated strong accuracy and effectively mitigated the overfitting problem in canola organ segmentation.

We compared five representative state-of-the-art (SOTA) models for point cloud semantic segmentation: SPG [46], PointMetaBase [47], PointVector [48], PointNext [49], and PointNet++ [17]. Except for PointNet++, which has a relatively simple architecture, the other models have multiple variants. To fairly evaluate the upper-bound performance of each model, we used the best-performing official version for all comparisons. SPG adopts a dual-branch structure, with the main branch selectable among PointNet++, PTV1 [50], and PTV2 [51]. In this study, we selected the strongest version, PTV2, as the main branch. The parameter settings of all comparison models followed the recommendations of their original papers.

As shown in Table 4, KAN-GLNet achieved the highest segmentation accuracy among all compared models, demonstrating outstanding segmentation capability. Specifically, it achieved 94.50% mIoU, 96.72% mAcc, and 97.77% OAcc, with only 5.72M parameters, which are 4.46, 2.43, and 1.72 percentage points higher than those of the second-best model, SPG, respectively, while using only 50.6% of its parameter count. In addition, compared to other models such as PointMetaBase and PointVector, KAN-GLNet not only achieved significantly higher segmentation accuracy but also had a much smaller model size. Overall, KAN-GLNet achieves the highest segmentation accuracy with a relatively compact model, effectively addressing the trade-off between model size and accuracy in plant point cloud segmentation.

Fig 9 shows the visualized segmentation results of six models on the test dataset. Despite the dense spatial arrangement of siliques, KAN-GLNet was able to accurately distinguish individual siliques, especially showing strong performance in identifying the boundaries between siliques and stems. As an improved Transformer-based model, SPG also performed

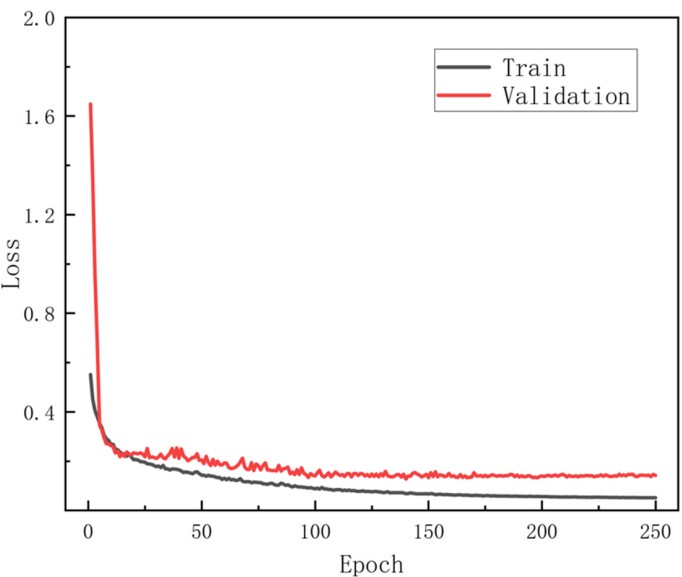
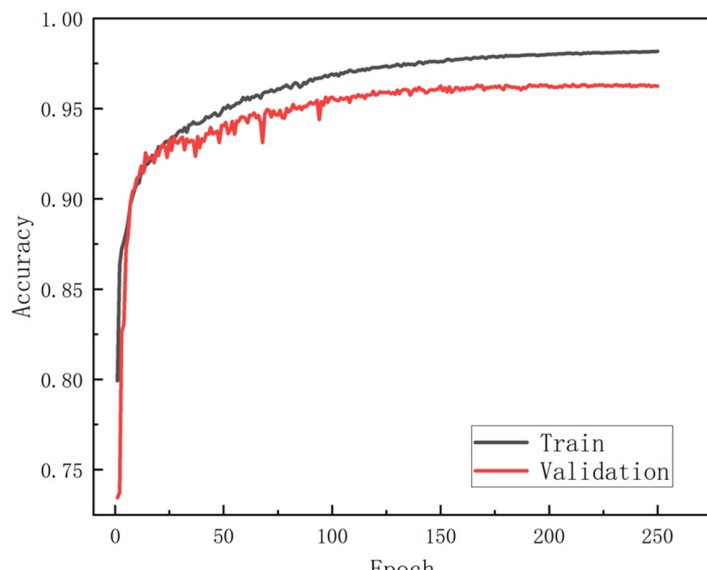

**Fig 8**. Training loss and accuracy curves of KAN-GLNet.

**Table 4**. A comparison of semantic segmentation performance across five networks, with the top results highlighted in bold.

| Method | mIoU(%) | mAcc(%) | OAcc(%) | silique(%) | stem(%) | Params(M) |
|---|---|---|---|---|---|---|
| PointNet++ | 70.40 | 77.68 | 87.76 | 85.6 | 55.2 | 0.97 |
| PointNext | 75.55 | 82.69 | 89.61 | 87.28 | 63.82 | 41.58 |
| PointVector | 82.74 | 88.62 | 92.71 | 90.70 | 72.49 | 24.10 |
| PointMetabase | 83.50 | 89.00 | 93.08 | 91.16 | 75.83 | 24.06 |
| SPG | 90.04 | 94.29 | 95.85 | 94.49 | 85.59 | 11.3 |
| **KAN-GLNet(Our)** | **94.50** | **96.72** | **97.77** | **97.0** | **92.0** | 5.72 |

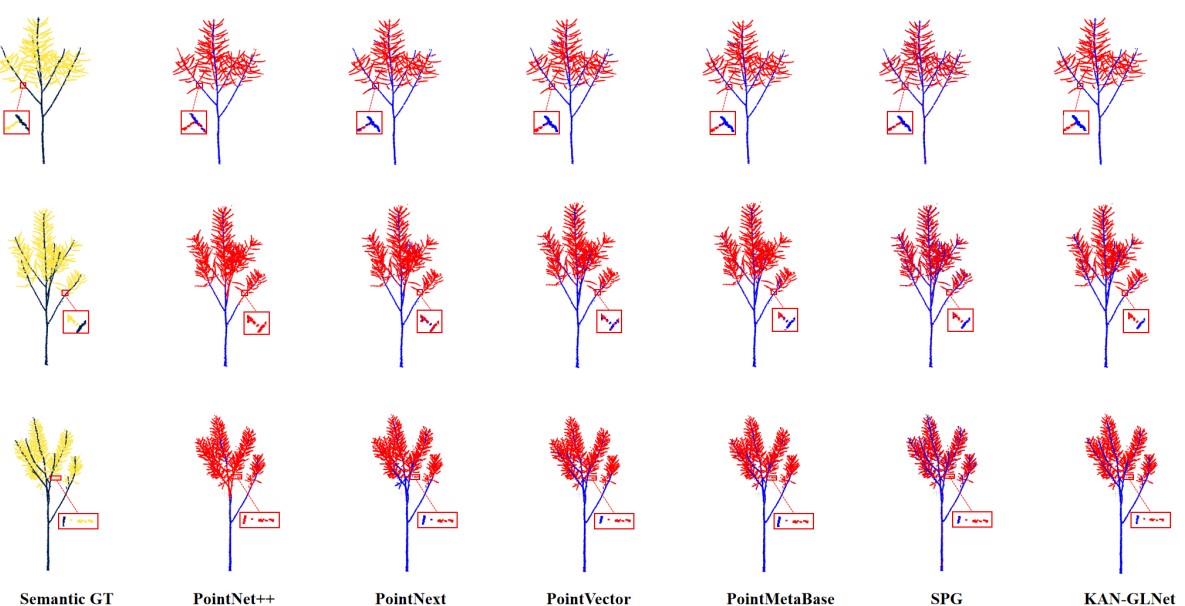

| Semantic GT | PointNet++ | PointNext | PointVector | PointMetaBase | SPG | KAN-GLNet |

**Fig 9**. **KAN-GLNet and baseline models segmentation visualization on test set.**

well in boundary distinction, but it often misclassified large regions of siliques or stems into the wrong category, leading to semantic confusion. PointMetaBase and PointVector exhibited similar performance. Their ability to distinguish boundaries in densely clustered siliques was limited, often misidentifying upper stems as siliques, although their accuracy improved in regions with sparse silique distribution. PointNext tended to misclassify stems as siliques over large areas in dense regions and also struggled with accuracy in sparse regions.

## Canola silique clustering experiment

The DBSCAN algorithm relies on two critical hyperparameters. The neighborhood radius eps defines the maximum distance within which points are considered neighbors, while the minimum number of samples min_samples specifies the minimum number of neighboring points required for a point to be classified as a core point. In this study, we selected eps = 0.10 and min_samples = 5 as the final parameter combination for silique instance counting in canola plants.

We compared the clustering results with manual measurements to evaluate the accuracy of the improved DBSCAN algorithm in instance-level silique counting. As shown in Table 5, the silique counts for individual canola plants exhibited strong consistency. The overall evaluation metrics were as follows: a Mean Absolute Error (MAE) of 4.80, a Root Mean Square Error (RMSE) of 5.25, and a Counting Accuracy (CA) of 97.45%. These three metrics respectively validate the reliability of the clustering results in terms of error magnitude, prediction stability, and overall accuracy, indicating that the improved method can effectively reflect the actual number of plant organs.

**Table 5.** Comparison between the actual number and the detected number of canola siliques.

| Plant number | Actual silique number | Predicted silique number | Error count | CA(%) |
|---|---|---|---|---|
| 1 | 78 | 75 | 3 | 96.15 |
| 2 | 144 | 142 | 2 | 98.61 |
| 3 | 293 | 285 | 8 | 97.27 |
| 4 | 228 | 222 | 6 | 97.37 |
| 5 | 232 | 227 | 5 | 97.84 |
| **Total** | 975 | 951 | 24 | 97.45 |

Building on the original DBSCAN clustering process, we introduced SOR filtering to remove local noise introduced during the semantic segmentation stage, thereby improving the purity of the point cloud input. This enhancement significantly increases the robustness of DBSCAN in densely entangled regions, enabling it to accurately separate neighboring organs even under severe clustering, occlusion, and overlap of siliques. The results demonstrate that our proposed clustering pipeline is not only simple and efficient but also well-suited to the high-precision and high-adaptability demands of organ-level recognition in plant phenotyping.

## Ablation experiment

In this study, we systematically evaluated the individual and combined effects of three key modules: Reverse Bottleneck KAN Convolutions, GLFN, and ContraNorm. These modules were integrated into the baseline PointNet++ model both independently and jointly for thorough validation. Table 6 presents a comparison of segmentation performance across different module combinations on the NeRF canola point cloud dataset.

The experimental results show that introducing the Reverse Bottleneck KAN Convolutions leads to the most significant improvement in segmentation performance. This demonstrates that the module is highly effective in extracting spatial structural features, enabling the network to better recognize complex geometric boundaries such as the junctions between siliques and stems, while maintaining a relatively low number of parameters. The GLFN module enhances the model's ability to perceive spatial details by fusing multi-scale global and local contextual information. ContraNorm improves the overall discriminative capability by introducing a contrastive learning mechanism that alleviates feature degradation and reduces overlap between different classes.

It is worth noting that when all three modules are combined to form the complete KAN-GLNet architecture, the model achieves the highest performance in terms of mean Intersection over Union and overall accuracy, indicating improved segmentation precision and stronger recognition of dominant structural components. However, the mean class accuracy is slightly lower than that of the model without the GLFN module. This is likely because the adaptive modulation in

**Table 6.** Ablation experiments on KAN-GLNet were conducted using different modules and combinations on the testing set.

| ContraNorm | GLFN | Reverse Bottleneck KAN | mIoU(%) | mAcc(%) | OAcc(%) | Params(M) |
|---|---|---|---|---|---|---|
| ✓ | | | 73.80 | 80.41 | 89.18 | 0.97 |
| | ✓ | | 78.18 | 83.90 | 91.03 | 1.85 |
| | | ✓ | 93.50 | 96.27 | 97.35 | 4.83 |
| ✓ | ✓ | | 79.18 | 84.97 | 91.37 | 1.85 |
| ✓ | | ✓ | 93.88 | **96.74** | 97.49 | 4.83 |
| | ✓ | ✓ | 94.29 | 96.72 | 97.68 | 5.72 |
| ✓ | ✓ | ✓ | **94.50** | 96.72 | **97.77** | 5.72 |

The presence of the ✓ symbol denotes utilization of the corresponding enhanced module at that specific position, whereas its absence signifies retention of the original architectural components from the baseline model.

GLFN strengthens the fusion of global and local features, but at the same time introduces feature bias, which reduces the model's attention to minority class samples and ultimately results in a slight drop in mean class accuracy.

## Convolution comparison

In this section, we select PointNet++ as the baseline model. To reduce the spatial dimensionality of features, we replace the first convolution layer in the SA module with Kolmogorov–Arnold Convolutions, FastKAN Convolutions, and Reverse Bottleneck KAN Convolutions, with all convolution kernels set to size 3. Table 7 presents the comparison of segmentation accuracy and parameter count for these improved models on the NeRF canola point cloud dataset.

Experimental results show that introducing Kolmogorov–Arnold series convolutions significantly improves the model's segmentation performance for canola siliques and stems. However, both Kolmogorov–Arnold Convolutions and replacing only the basis functions lead to a substantial increase in parameter count, limiting the model's application in resource-constrained environments. Given that this study aims to maintain segmentation accuracy while minimizing model complexity, we designed and introduced Reverse Bottleneck KAN Convolutions. The results indicate that this convolution not only slightly outperforms the model using FastKAN Convolutions in segmentation accuracy but also significantly reduces the parameter count, demonstrating an excellent balance between low parameter count and high precision and showing stronger potential for practical applications.

## Convolution replacement position

This paper introduces the Reverse Bottleneck KAN Convolution and applies it to the Min-PointNet part of the PointNet++ model by replacing the first convolutional layer, resulting in the design of KGL-PointNet. Table 8 presents the impact of replacing convolution layers at shallow, deep, and multiple levels on model performance and parameter count. Experimental results show that when the Reverse Bottleneck KAN Convolution is placed at the shallowest layer of the feature extraction network, the model achieves the highest mIoU, mAcc, and OAcc, with only a slight increase in parameters compared to replacing the second layer. This indicates that shallow layers are more effective at capturing low-level geometric features and local texture information in the point cloud, while deeper layers focus more on semantic-level representations, where the advantages of the Reverse Bottleneck KAN Convolution become less pronounced.

Moreover, stacking multiple Reverse Bottleneck KAN Convolutions does not further improve segmentation performance and instead leads to a decline. The primary reason lies in the significant increase in parameter count, which introduces

**Table 7**. **Comparison of segmentation accuracy and parameter count between Kolmogorov-Arnold convolutions, FastKAN convolutions, reverse bottleneck KAN convolutions, and the baseline model.**

| Model | mIoU(%) | mAcc(%) | OAcc(%) | Params(M) |
|---|---|---|---|---|
| Conv (baseline) | 70.40 | 77.68 | 87.76 | 0.97 |
| Kolmogorov-Arnold Convolutions | 92.58 | 95.67 | 96.96 | 7.98 |
| FastKAN Convolutions | 93.20 | 95.83 | 97.23 | 7.98 |
| **Reverse Bottleneck KAN Convolutions** | **93.50** | **96.27** | **97.35** | 4.83 |

**Table 8**. **Comparison of reverse bottleneck KAN convolutions replacing single-layer and multi-layer convolutions (Conv).**

| Conv1 | Conv2 | Conv3 | mIoU(%) | mAcc(%) | OAcc(%) | Params(M) |
|---|---|---|---|---|---|---|
| ✓ | | | **93.50** | **96.27** | **97.35** | 4.83 |
| | ✓ | | 91.05 | 94.55 | 96.32 | **4.80** |
| | | ✓ | 88.39 | 92.97 | 95.15 | 8.28 |
| ✓ | ✓ | | 93.21 | 96.00 | 97.22 | 8.66 |
| ✓ | ✓ | ✓ | 92.22 | 95.18 | 96.83 | 15.97 |

Conv1–3 are Min-PointNet's layers 1–3. ✓ marks indicate Reverse Bottleneck KAN convolutions; unmarked layers remain original.

risks of overfitting and increased computational burden. On the one hand, a large number of parameters makes the network more prone to falling into local minima; on the other hand, excessive feature enhancement may result in unstable gradient updates or redundant information, thereby weakening the model's generalization ability. Therefore, the optimal strategy is not to simply increase the number of modules but to analyze the importance of features at different layers in depth, adopting selective replacement or dynamic configuration methods to maximize the effectiveness of shallow-layer geometric feature extraction while maintaining model compactness.

## DBSCAN optimal parameter search and SOR filter necessity

To determine the optimal DBSCAN parameter combination suitable for the clustering task of canola siliques, this study adopts a grid search method to systematically evaluate the impact of two key parameters—neighborhood radius (eps) and minimum number of samples (min_samples)—on clustering performance. The parameter search range was set as eps ∈ [0.05, 0.20] and min_samples ∈ [1, 15]. As shown in Fig 10, the experimental results indicate that when the parameter combination is set to eps = 0.10 and min_samples = 5, the optimized DBSCAN demonstrates the best clustering performance in regions where the density of siliques varies significantly.

The study found that a smaller eps value effectively distinguishes neighboring silique clusters that are spatially close, thereby avoiding under-segmentation. However, due to uneven point cloud density, it may also lead to some siliques being misclassified as noise. On the other hand, a moderate min_samples value not only suppresses over-segmentation caused by noise or residual points from semantic segmentation but also avoids mistakenly discarding true silique points when the threshold is set too high.

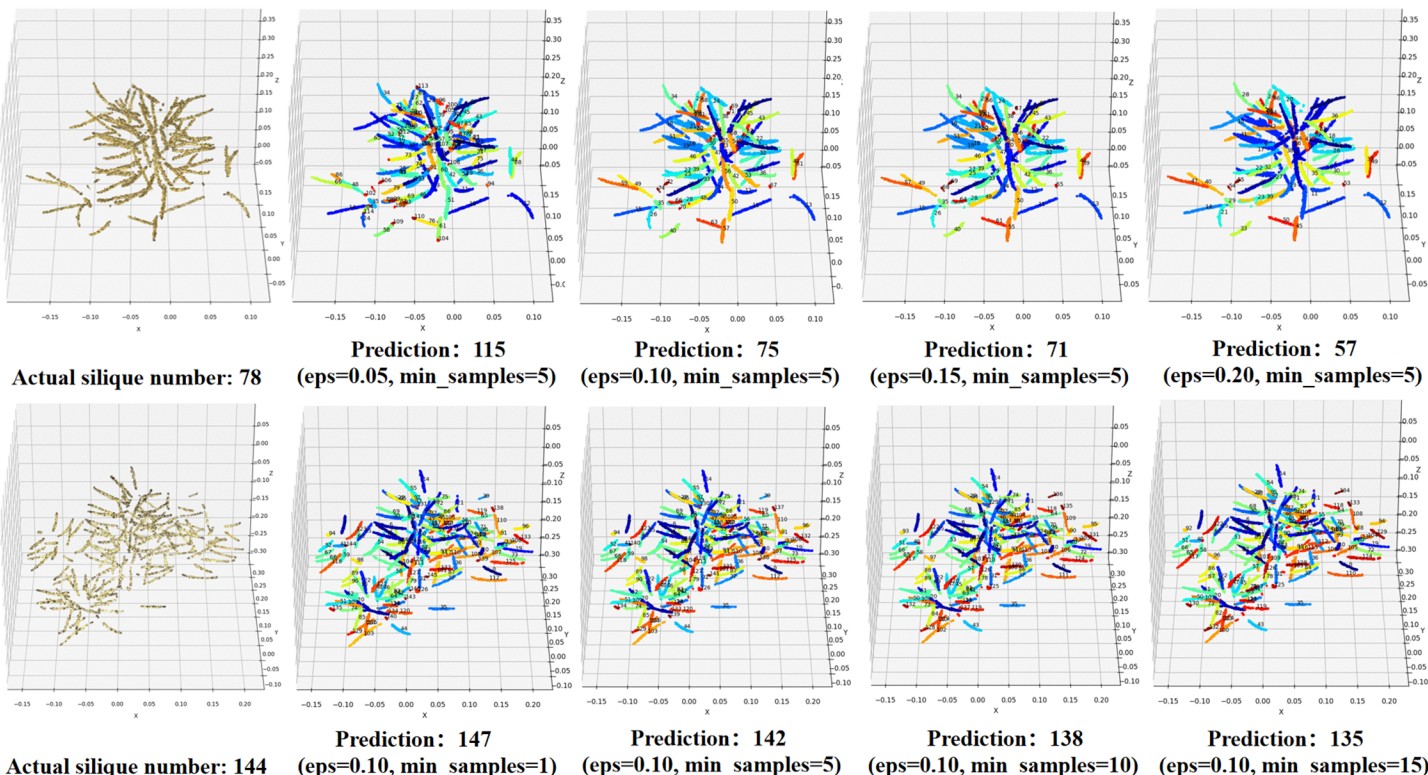

Fig 10. **DBSCAN hyperparameter search.** (A) Impact of varying eps parameters on clustering performance (ground truth silique count: 78). (B) Impact of varying min_samples parameters on clustering performance (ground truth silique count: 144).

Since raw silique point clouds often contain acquisition noise and segmentation errors, directly applying the standard DBSCAN tends to misclassify these outliers as independent clusters, resulting in the generation of false clusters and over-segmentation. This negatively impacts the stability and repeatability of subsequent instance counting and morphological analysis. As shown in Fig 11, standard DBSCAN often counts outliers as false clusters, whereas the improved method effectively addresses this issue.

By introducing statistical outlier removal (SOR) filtering into the clustering pipeline, we first eliminate the most sparsely distributed points caused by segmentation residuals and acquisition noise, thereby making silique cluster boundaries clearer and enhancing their connectivity. Subsequently, using the optimized parameters of eps = 0.10 and min_samples = 5, clustering is performed on the remaining point cloud. This allows for an adaptive balance between intra-cluster connectivity and inter-cluster separation in both dense and sparse regions.

As a result, the generation of false small clusters is significantly reduced, and each real silique cluster can be accurately extracted, greatly enhancing the robustness and consistency of the clustering results. Ultimately, this improved pipeline proves particularly effective in extracting phenotypic parameters from plants with complex and overlapping structures such as fruits and leaves. It provides a reliable data foundation for high-throughput quantification of canola siliques, reduces the burden of manual correction, and offers solid algorithmic support for subsequent three-dimensional organ morphological studies.

## Discussion

### Multi-view point cloud acquisition and generation

We designed a dedicated 3D scanning system for plants aimed at efficiently acquiring multi-view point cloud data. The system fully considers the morphological characteristics of plants such as canola and supports continuous detail capture from multiple angles. Specifically, canola plants are placed on a low-speed rotating turntable, and the system uses fixed cameras to record video of the entire rotation process, avoiding mechanical interference and timing errors caused by frequent photographing. This method allows comprehensive acquisition of the plant's 3D appearance without affecting its

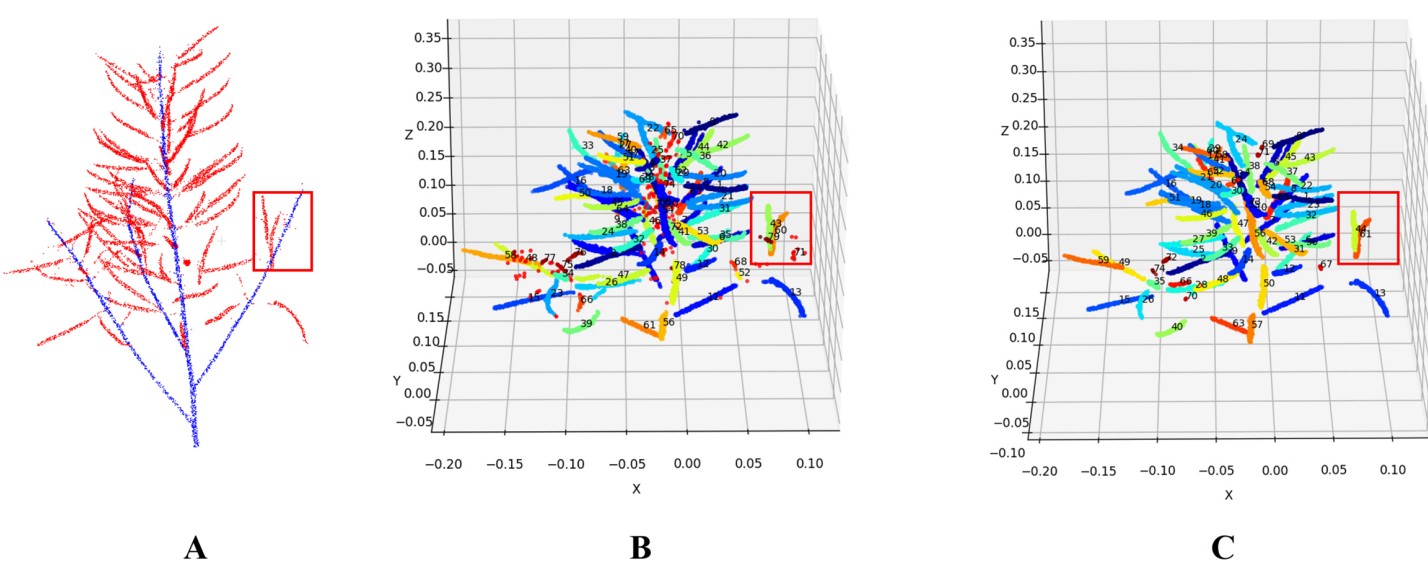

**A**    **B**    **C**

**Fig 11. Comparison of DBSCAN clustering with and without SOR filtering.** (A) Original silique labels. (B) Clustering result using DBSCAN. (C) Clustering result using DBSCAN with SOR filtering.

structure. Subsequently, the video is processed frame-by-frame using the ffmpeg tool to obtain continuous, complete, and multi-angle image sequences, providing sufficient data for point cloud reconstruction.

During the shooting process, the system employs stable and continuous lighting to eliminate shadow interference and maintain uniform illumination. At the same time, a solid color background is used to effectively reduce background noise, further enhancing image quality and laying a solid foundation for point cloud generation.

For point cloud generation, we adopted NeRF technology, which can rapidly generate point cloud data with clear structure and high accuracy. Combined with the front-end video acquisition and image processing workflow, NeRF significantly improves the efficiency and quality of reconstruction, providing high-quality foundational data for subsequent semantic segmentation and clustering analysis.

## Analysis of experimental results

During the silique maturation stage, canola siliques are arranged in a disorderly and heavily overlapping manner, while the number of branches varies significantly, posing considerable challenges for segmentation. KAN-GLNet effectively addresses these difficulties, successfully achieving precise segmentation, and the subsequently optimized DBSCAN algorithm effectively resolves the issue of numerous silique outliers being regarded as false clusters, providing a solid foundation for accurate counting and phenotypic analysis.

In the field of plant point cloud semantic segmentation, commonly used models are based either on point-based architectures or transformer architectures, both of which struggle to achieve an excellent balance between model accuracy and parameter size. For example, Dong et al. [20] proposed a segmentation model with a relatively low parameter count of 6.08M, achieving stem and leaf segmentation accuracies of 89.21% mIoU on sugarcane, 89.19% mIoU on maize, and 83.05% mIoU on tomato, demonstrating the potential of lightweight design. However, the relatively lower segmentation accuracy limits the precision of subsequent fine-grained phenotypic extraction. On the other hand, Ma et al. [22] proposed PSTNet, a transformer-based model that achieved 92.20% IoU on eggplant point clouds, but its large parameter size results in high computational cost and deployment difficulty, which is unfavorable for resource-constrained practical scenarios.

Our proposed model KAN-GLNet achieved 94.50% mIoU, 96.72% mAcc, and 97.77% OAcc on the NeRF canola test set, significantly outperforming the second-place SPG model based on PTV2 in terms of accuracy. Meanwhile, our model has only 5.72M parameters, which is far fewer than most Transformer-based models, reflecting an excellent balance between model accuracy and parameter size. This not only ensures high segmentation accuracy but also greatly reduces the computational resource requirements of the model, providing an efficient point cloud processing solution for agricultural edge devices.

## Limitations

In this study, we acquired point clouds of canola plants using a self-designed multi-view image acquisition platform and applied KAN-GLNet for segmentation and clustering to extract silique counts. However, this process still faces two major limitations that need to be addressed.

As shown in Fig 12, canola plants growing naturally in field plots are dense and crowded, with severe silique overlap, making it difficult to capture complete multi-view images without damaging the overall plant structure. Although we transplanted individual plants into pots for imaging and used a low-speed turntable and uniform lighting to ensure data quality to some extent, this process inevitably altered the natural growth state of branches and leaves. As a result, the acquired point clouds differ from those in real field conditions, affecting the accuracy of segmentation and clustering outcomes in reflecting phenotypic traits under natural environments. Future work could explore non-contact aerial or multi-angle scanning technologies such as drones or mobile robotic arms to minimize plant structure interference. At the same time, future work could also shift the focus from individual canola plants to large-scale field-level phenotypic estimation.

 

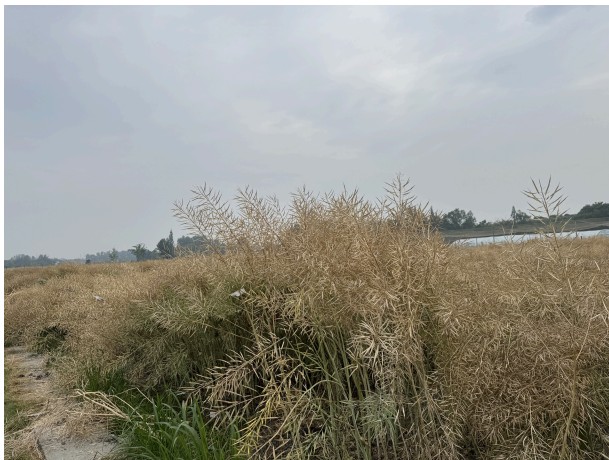

**Fig 12**. **Canola experimental field at Sichuan Agricultural University, Ya'an City, Sichuan Province, China.**

Our designed KAN-GLNet achieved 94.50% segmentation accuracy on the NeRF canola dataset, a significant improvement over the baseline PointNet++, which achieved only 70.40%, and it also outperformed other SOTA models. However, this improvement came with a parameter increase from the baseline's 1M to 5.7M, posing challenges for deployment on resource-constrained edge devices or in real-time processing scenarios. Future work could further investigate model compression and lightweight optimization techniques, such as weight quantization, channel pruning, or knowledge distillation, to reduce model complexity while maintaining segmentation performance, thereby providing a viable pathway for applications in agricultural drone inspection, intelligent greenhouses, and mobile terminals.

This study also has certain limitations in the application of data augmentation strategies. Although the static augmentation methods employed (such as rotation, scaling, and random dropout) have been widely validated in the field of point cloud processing and can effectively simulate real-world disturbances such as pose variations and occlusions during field data acquisition, the diversity of the generated data remains constrained by the predefined transformation space, making it difficult to cover the complex variations present in all real scenarios. Moreover, to evaluate the robustness of the model when facing input perturbations, this study also applied data augmentation to the validation and test sets. While this design improved the stability of statistical evaluation under small-sample conditions and allowed a sharper focus on answering the robustness question of "whether the model performs consistently under real disturbances," the results may not be directly comparable to traditional generalization performance evaluations based solely on original data. Future work could further introduce dynamic augmentation techniques and also provide evaluation results based on the original test set, so as to more comprehensively reflect the performance of the model.

## Conclusion

In summary, this study utilized NeRF technology to generate accurate point clouds of canola plants and proposed a novel method for canola point cloud segmentation and silique counting based on KAN-GLNet and an optimized DBSCAN algorithm.

KAN-GLNet was developed based on the PointNet++ model and includes three major improvements: Reverse Bottleneck KAN Convolution, the GLFN feature modulation block, and ContraNorm. The Reverse Bottleneck KAN Convolution is used to enhance feature extraction capability, the GLFN feature modulation block is designed to optimize the

fusion of local and global information, and ContraNorm, based on contrastive learning, is introduced to prevent over-smoothing. Experimental results show that KAN-GLNet outperforms multiple advanced models in canola semantic segmentation tasks, achieving 94.50% mIoU, 96.72% mAcc, and 97.77% OAcc, with a model parameter size of only 5.72M. This demonstrates that KAN-GLNet achieves an excellent balance between low parameter size and high accuracy, showing strong practical potential and providing a feasible and efficient technical solution for applications in agriculture, especially in resource-constrained environments such as edge devices, drone inspections, and mobile terminals.

Future research will focus more on phenotypic studies of canola in open field conditions, aiming to achieve the transition and expansion from single-plant to population-scale analysis. Given the dense distribution of plants, severe silique occlusion, and complex lighting conditions in natural environments, subsequent work could explore non-contact high-throughput acquisition methods in complex field scenarios, such as integrating drone-based aerial photography and multi-angle robotic arm scanning to obtain large-scale point cloud data under real-world conditions. To further improve the real-time performance and accuracy of field data processing, it is necessary to optimize the model architecture, introduce more efficient lightweight strategies, or incorporate multimodal information such as RGB images and depth data to enhance the model's robustness and generalization ability. In addition to silique counting, future work could be extended to the accurate extraction of more canola phenotypic traits, such as silique length, stem diameter, and plant height, providing more comprehensive data support for canola phenotyping research and intelligent breeding.

## Author contributions

**Conceptualization:** Jiajun Liu.

**Data curation:** Jie Liu.

**Funding acquisition:** Jie Liu.

**Investigation:** Di Hu.

**Methodology:** Jiajun Liu.

**Project administration:** Junjie Wu.

**Supervision:** Bei Zhou, Jiangshu Wei.

**Validation:** Jiajun Liu, Xike Zhang, Yao Zhang.

**Visualization:** Jiajun Liu, Changping Wu.

**Writing – original draft:** Jiajun Liu.

**Writing – review & editing:** Jiajun Liu, Bei Zhou.

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
