## [Decision Letter · Decision Letter 0]

19 May 2025

PONE-D-25-17533KAN-GLNet: An Enhanced PointNet++ Model for Canola Silique Segmentation and CountingPLOS ONE

Dear Dr. Zhou,

Thank you for submitting your manuscript to PLOS ONE. After careful consideration, we feel that it has merit but does not fully meet PLOS ONE’s publication criteria as it currently stands. Therefore, we invite you to submit a revised version of the manuscript that addresses the points raised during the review process. Specifically, this work has only few images with large number of parameters, which is not proper for deep learning algorithm. Please address this question clearly.

We look forward to receiving your revised manuscript.

Kind regards,

Xiaoyong Sun

Academic Editor

PLOS ONE

 [This project is supported by National Natural Science Foundation of China, grant number 32301762.]. 

Additional Editor Comments (if provided):

Reviewers' comments:

Reviewer's Responses to Questions

**Comments to the Author**

1. Is the manuscript technically sound, and do the data support the conclusions?

Reviewer #1: Yes

Reviewer #2: Partly

Reviewer #3: Partly

2. Has the statistical analysis been performed appropriately and rigorously?

Reviewer #1: Yes

Reviewer #2: No

Reviewer #3: No

3. Have the authors made all data underlying the findings in their manuscript fully available?

Reviewer #1: No

Reviewer #2: No

Reviewer #3: Yes

4. Is the manuscript presented in an intelligible fashion and written in standard English?

Reviewer #1: Yes

Reviewer #2: No

Reviewer #3: Yes

5. Review Comments to the Author

Reviewer #1: 1. The full forms of KAN-GLNet, DBSCAN, antd GLFN should be included at least once in the abstract, preferably at the beginning.

2. The introduction needs to be broadened by citing similar previous studies involving the algorithm in agriculture; additionally, the objectives should be framed based on the limitations of these related studies.

3. Experimental results should not be included within the objectives of the study.

4. The source of image acquisition must be clearly stated, as it is critical to dataset quality. Specifications should be tabulated for better readability. Further, both natural and artificial environmental conditions used for dataset creation must be elaborated. The following reference should be cited for improving dataset acquisition presentation.

Paul, A., Machavaram, R., Kumar, D., & Nagar, H. (2024). Smart solutions for capsicum Harvesting: Unleashing the power of YOLO for Detection, Segmentation, growth stage Classification, Counting, and real-time mobile identification. Computers and Electronics in Agriculture, 219, 108832.

5. NeRFStudio must be properly cited using an appropriate and complete citation format.

6. The term "4.4-fold expansion" in dataset processing must be technically described, indicating the specific augmentation or data multiplication steps applied.

7. Equation numbers must be cited properly in the text, similar to how figures and tables are referenced.

8. The dataset division into only training and testing sets is inadequate. The absence of a validation set is a limitation. The rationale behind choosing 180 training points and 40 test points should be justified. Additionally, methods like cross-validation can be used to obtain more reliable performance estimates.

9. Software and hardware specifications used for training should be presented in a table to enhance clarity and readability.

10. The choice of 300 training epochs must be justified based on the convergence behavior of the loss curves. Include loss curve plots to illustrate the training dynamics. Also, provide the rationale for choosing a batch size of 8 and a learning rate of 0.01.

11. Results shown in Table 4 and Table 5 must be discussed in detail within the text, elaborating on the significance and implications of the values presented.

12. Figure 10 is blurry and lacks clarity; it should be reworked for improved visibility and resolution.

13. The results of the current study should be critically compared with outcomes from similar previous studies to contextualize its contributions.

14. The limitations of the study should be clearly outlined in the second paragraph of the conclusion to offer a balanced perspective on the work.

Reviewer #2: While the proposed method demonstrates promising segmentation and counting capabilities, the quantitative result visualizations in the current form lack clarity and are difficult to interpret. Moreover, the reliance on the DBSCAN clustering algorithm for instance separation raises concerns about its robustness across diverse plant samples, particularly in densely packed or overlapping regions. It is unclear whether DBSCAN consistently yields correct instance counts for all cases. To strengthen the work, the authors should provide clearer, high-resolution visualization of results and include statistical metrics—such as mean absolute error (MAE), root mean squared error (RMSE), and count accuracy—to quantitatively validate the reliability of the predicted instance counts across the dataset. This would provide a more rigorous and interpretable evaluation of their approach.

Reviewer #3: The manuscript introduces a modified model architecture for pointcloud segmentation tasks. While the proposed model is well described and the authors provide detailed explanations, several important concerns limit the strength of the conclusions and the reliability of the results.

Dataset Size vs. Model Complexity: The model contains over 5 million parameters but is trained on a dataset consisting of only 50 original samples, augmented to a total of 220. This is a very small dataset for such a large model, and the manuscript does not provide evidence that steps were taken to mitigate overfitting. Notably, there is no mention of a validation set being used during training, which would be important for monitoring model generalization and preventing overfitting.

Potential Data Leakage: The test set is selected from the same pool of 220 samples, of which 170 were generated through data augmentation. Since only 50 original images exist, there is a high likelihood that augmented versions of the same original images may have been used across both the training and test sets. This raises concerns about potential data leakage, which could artificially inflate the reported performance.

Although the proposed model may offer value, the experimental setup makes it difficult to assess its true effectiveness. Addressing the issues of overfitting and potential data leakage will be critical to strengthen the validity of the results.

6. PLOS authors have the option to publish the peer review history of their article (what does this mean?). If published, this will include your full peer review and any attached files.

Reviewer #1: **Yes: **Ayan Paul

Reviewer #2: No

Reviewer #3: No

---

## [Author Response · Author response to Decision Letter 1]

20 Jul 2025

PLOS ONE

稿件编号�PONE-D-25-17533

KAN-GLNet: An Enhanced PointNet++ Model for Canola Silique Segmentation and Counting

Jiajun Liu1 , Bei Zhou1,2*, Jie Liu3 , Xike Zhang1 , Jiangshu Wei1,2, Yao Zhang1 , Junjie

Wu1 , Changping Wu3 , Di Hu1

1 College of Information Engineering, Sichuan Agricultural University, Yaan 625000, China

2 Sichuan Key Laboratory of Agricultural Information Engineering, Sichuan Agricultural University, Yaan 625000, China

3 College of Agronomy, Sichuan Agricultural University, Chengdu 610000, China

* Corresponding author: 12801@sicau.edu.cn

Based on the suggestions of several reviewers, we not only addressed each of the specific concerns they raised, but more importantly, we implemented a more rigorous division of the original dataset and conducted all experiments anew accordingly.

① In our initial experiments, we expanded 50 original point clouds to 220 samples using data augmentation techniques. These were then split into training and testing sets for model training. The model demonstrated good fitting behavior during training and achieved high segmentation accuracy for canola siliques, with a mean Intersection over Union (mIoU) of 90.20%. In the subsequent clustering stage, the DBSCAN algorithm enabled us to achieve over 95% accuracy in silique counting. However, this experimental setup raised concerns from reviewers, mainly regarding the limited dataset size, the absence of a validation set (making it difficult to assess generalization and potential overfitting), and the risk of data leakage potentially inflating model performance.

② In response, we first divided the dataset into training, validation, and testing sets in a 7:2:1 ratio. We then applied data augmentation independently within each subset, expanding the dataset by a factor of 10, resulting in a total of 550 point clouds. Additionally, in the semantic segmentation stage, we introduced more recent state-of-the-art (SOTA) models such as SPG (2024) and PointMetabase (2023) for a comprehensive performance comparison. The results showed that during training, the loss curves for both training and validation sets exhibited good convergence, addressing concerns about the model’s generalization ability and overfitting. Furthermore, among all baseline models, ours achieved the highest segmentation accuracy (mIoU of 94.5%) with a relatively small number of parameters, fully demonstrating the superiority of our approach. Finally, in the clustering stage, we proposed a novel DBSCAN processing pipeline that more accurately identified individual siliques, achieving a silique recognition accuracy of 97.42%. These results underscore the advantages of our method and offer a promising new solution for agricultural scenarios, especially in resource-limited settings.

③In the revised manuscript, we will use the following formatting: reviewer comments in black italics, our responses in black regular font, and revisions in the manuscript in blue regular font, ensuring clear distinction.

Response to Editor

Comment 1: Specifically, this work has only few images.

Authors Response:

Dear Editor,

We sincerely appreciate your valuable feedback. Regarding the concern about limited image data, please allow us to provide a detailed clarification:

① Challenges in 3D Point Cloud Acquisition

Acquiring 3D point clouds is inherently more complex than capturing 2D images due to two key factors: the requirement for specialized sensors and data collection protocols, and the extensive manual annotation needed for intricate plant structures (Ma et al., Remote Sensing, 2019). For context, each rapeseed plant point cloud in our dataset was reconstructed from 120 images, meaning the 50-plant dataset effectively represents approximately 6,000 raw images.

② Limited Scale of Public Plant Point Cloud Datasets

Existing public datasets in this domain typically contain modest sample sizes. For example: ROSE-X consists of only 11 annotated rose bush models; Soybean-MVS includes 102 point clouds covering 5 soybean varieties across 13 growth stages; Pheno4D contains 204 point clouds (84 maize, 140 tomato); and PLANesT-3D has merely 34 point clouds (10 pepper, 10 rose, 14 ribes).

③ Proven Feasibility of Small-Scale Training

Prior research has successfully demonstrated plant phenotyping with limited data. Guo et al. (2023) achieved accurate leaf counting using only 60 augmented cabbage point clouds combined with an improved DBSCAN algorithm. Similarly, Ma et al. obtained good model convergence with just 50 eggplant seedling point clouds, confirming that meaningful analysis is possible even with constrained datasets.

④ Substantial Dataset Expansion in Revision

In response to your comments, we have significantly expanded our dataset to 550 point clouds (a 10-fold increase from the original), while rigorously preventing data leakage. Our models demonstrate excellent training/validation loss convergence and outperform all state-of-the-art benchmarks in accuracy metrics.

Comment 2�Large number of parameters, which is not proper for deep learning algorithm. 

Authors Response: We fully understand you are concern on this issue and sincerely appreciate this valuable comment. Below is our detailed explanation regarding this matter:

① Model Innovations

The core contribution of our work is the KAN-GLNet model, which introduces three key innovations: an inverted KAN convolution for enhanced feature extraction, a GLFN block for effective multiscale local-global information fusion, and a ContralNormal module that improves feature stability through contrastive learning. The model achieves superior performance (94.50% mIoU, 96.72% mAcc, 97.77% OAcc) compared to all baseline methods.

② Domain-Specific Methodological Requirements

In plant phenotyping research, deep learning segmentation alone is often insufficient. As established by Ferrara et al. (2018), downstream clustering algorithms like DBSCAN remain essential for organ-level analysis. Our work addresses this by: developing an optimized DBSCAN workflow with systematic parameter selection guidelines, and providing thorough evaluation of its application in silique clustering tasks.

We firmly believe this work not only meets deep learning research standards but also makes meaningful contributions to 3D plant phenotyping. We are deeply grateful for the editorial team's expertise and constructive suggestions, which have significantly improved our manuscript.

Comment 3: Please ensure that your manuscript meets PLOS ONE's style requirements, including those for file naming.

Authors Response: Thank you for your attention to this matter. Our manuscript was prepared using LaTeX and submitted in PDF format. The LaTeX template we used was downloaded from the PLOS ONE official website at the following link: https://journals.plos.org/plosone/s/latex#loc-references. Additionally, we have carefully checked the formatting of figures, tables, references, and other elements to ensure compliance with the requirements. If you have any questions or identify any issues with our formatting, we sincerely appreciate your feedback so that we can make further detailed revisions.

Comment 4: Thank you for stating the following financial disclosure:

 [This project is supported by National Natural Science Foundation of China, grant number 32301762.].Please state what role the funders took in the study.  If the funders had no role, please state: "The funders had no role in study design, data collection and analysis, decision to publish, or preparation of the manuscript."If this statement is not correct you must amend it as needed.

Authors Response: Thank you for your attention to this work. We confirm that this project was supported by the National Natural Science Foundation of China, Grant No. 32301762, and that the funder participated in data collection and provided financial support.

Comment 5: Please include a separate caption for each figure in your manuscript.

Authors Response: We sincerely apologize for our oversight in the previous submission, where several figures were missing their corresponding titles. In our subsequent revisions, we have meticulously verified and ensured that every figure and table now has an appropriate title. Thank you very much for your understanding.

Response to Reviewer 1:

Comment 1: The full forms of KAN-GLNet, DBSCAN, antd GLFN should be included at least once in the abstract, preferably at the beginning.

Authors Response: Thank you for your valuable feedback. We sincerely appreciate bringing this oversight to our attention. The revised Abstract is now provided below in full, with all necessary corrections implemented.

Accurate analysis of plant phenotypic traits is crucial for crop breeding and precision

agriculture. This study proposes a lightweight semantic segmentation model named

KAN-GLNet(Kolmogorov–Arnold Network with Global–Local Feature Modulation),

based on an enhanced PointNet++ architecture and integrated with an optimized

Density-Based Spatial Clustering of Applications with Noise (DBSCAN) algorithm, to

achieve high-precision segmentation and automatic counting of canola siliques. A

multi-view point cloud acquisition platform was built, and high-fidelity canola point

clouds were reconstructed using Neural Radiance Fields (NeRF) technology. The

proposed model includes three key modules: Reverse Bottleneck Kolmogorov–Arnold

Network Convolution, a Global–Local Feature Modulation (GLFN) block, and a

contrastive learning-based normalization module called ContraNorm. KAN-GLNet

contains only 5.72M parameters and achieves 94.50% mIoU, 96.72% mAcc, and 97.77%

OAcc in semantic segmentation tasks, outperforming all baseline models. In addition,

the DBSCAN workflow was optimized, achieving a counting accuracy of 97.45% in the

instance segmentation task. This method achieves an excellent balance between

segmentation accuracy and model complexity, providing an efficient solution for

high-throughput plant phenotyping. The code and dataset have been made publicly

available at: https://anonymous.4open.science/r/KAN-GLNet-6432/.

Comment 2: The introduction needs to be broadened by citing similar previous studies involving the algorithm in agriculture; additionally, the objectives should be framed based on the limitations of these related studies.

Authors Response: We have carefully reviewed our previous manuscript and realized that the Introduction focused primarily on comparing our model's advantages while overlooking discussions of similar agricultural algorithms in prior studies. Accordingly, we have significantly expanded the Introduction to cover current research in this field, highlighting existing limitations that motivate our work. Specifically: Current plant phenotyping segmentation models struggle to balance performance and parameter efficiency, and we propose a novel point cloud segmentation model to address this trade-off. These revisions appear in Lines 40–73 of the revised manuscript.

To address these challenges, deep learning-based point cloud segmentation methods have emerged, automatically learning features through data-driven approaches. Early research primarily adopted voxelization to convert point clouds into structured data for feature extraction [13]. For example, Das Choudhury et al. [14] segmented maize stems and leaves using multi-view visual hull algorithms with voxel overlap verification and Euclidean clustering. Saeed et al. [15] implemented 3D segmentation of cotton stems, branches, and bolls via Point-Voxel Convolutional Neural Networks (PVCNN). However, these methods demand substantial computational resources and may incur information loss during segmentation. To overcome point cloud processing bottlenecks, subsequent research focused on direct point cloud processing architectures, mainly developing along two directions: point-based methods and Transformer-based approaches. The pioneering PointNet series [16, 17] laid the groundwork for point cloud deep learning, providing crucial technical support for plant 3D phenotyping. For instance, Ao et al. [18] applied PointNet for maize stem-leaf separation using local point density. Guo et al. [19] integrated PointNet++ with ASAP attention modules, achieving 86% mIoU in cabbage segmentation. Addressing PointNet++ optimization needs, PointNeXt achieved performance breakthroughs through training strategy updates and model scaling, with Dong et al. [20] reporting 89.21% mIoU (sugarcane), 89.19% mIoU (maize), and 83.05% mIoU (tomato) stem-leaf segmentation at 6.08M parameters. While such point-based models excel in parameter efficiency, they often face accuracy limitations due to insufficient feature extraction. PCT [21] pioneered Transformer architecture for 3D point cloud processing, marking its first application in point cloud segmentation. For example, Ma et al. [22] proposed PSTNet with cascaded self-attention (PSA) and local feature aggregation (NPA), achieving 92.20% IoU for eggplant point clouds. Yang et al. [23] developed PACANet, a Transformer-based pairwise attention center axis aggregation network, achieving 92.46% mean accuracy for maize populations at 46.2M parameters. These studies demonstrate that while Transformer architectures deliver breakthrough performance, they often suffer from parameter explosion, severely limitingdeployment in resource-constrained scenarios. To address the contradiction between parameter quantity and segmentation accuracy in plant point cloud segmentation, this study proposes KAN-GLNet, a

Kolmogorov-Arnold segmentation network equipped with global-local feature modulation. The model is designed to achieve high-precision segmentation under low-parameter constraints.

Comment 3: Experimental results should not be included within the objectives of the study.

Authors Response:Thank you for your correction. We have completed the replacement in the introduction section, specifically from lines 73 to 83, as shown below.

The key innovations include:

(i)Constructing the first NeRF-derived rapeseed silique point cloud dataset (50

samples), expanding sample scale through data augmentation strategies.

(ii)Proposing reverse bottleneck KAN convolution (enhanced Kolmogorov-Arnold

convolution) to replace conventional convolutions for strengthening feature extraction

capability.

(iii)Designing GLFN multi-scale feature modulation blocks to optimize spatial

representation of complex silique structures by fusing global-local attention mechanisms.

(iv)Introducing the ContraNorm contrastive normalization module to enhance point

cloud feature stability and segmentation consistency.

(v)Proposing an optimized DBSCAN workflow algorithm to achieve automated

silique instance counting.

Comment 4: The source of image acquisition must be clearly stated, as it is critical to dataset quality. Specifications should be tabulated for better readability. Further, both natural and artificial environmental conditions used for dataset creation must be elaborated. The following reference should be cited for improving dataset acquisition presentation. Paul, A., Machavaram, R., Kumar, D., & Nagar, H. (2024). Smart solutions for capsicum Harvesting: Unleashing the power of YOLO for Detection, Segmentation, growth stage Classification, Counting, and real-time mobile identification. Computers and Electronics in Agriculture, 219, 108832.

Authors Response:Thank you for bringing this issue to our attention. Our rapeseed was cultivated in the experimental fields of Sichuan Agricultural University

---

## [Decision Letter · Decision Letter 1]

29 Aug 2025

PONE-D-25-17533R1KAN-GLNet: An Enhanced PointNet++ Model for Canola Silique Segmentation and CountingPLOS ONE

Dear Dr. Zhou,

Thank you for submitting your manuscript to PLOS ONE. After careful consideration, we feel that it has merit but does not fully meet PLOS ONE’s publication criteria as it currently stands. Therefore, we invite you to submit a revised version of the manuscript that addresses the points raised during the review process. Please see the comments at the bottom of this email.

We look forward to receiving your revised manuscript.

Kind regards,

Xiaoyong Sun

Academic Editor

PLOS ONE

Journal Requirements:

Reviewers' comments:

Reviewer's Responses to Questions

**Comments to the Author**

1. If the authors have adequately addressed your comments raised in a previous round of review and you feel that this manuscript is now acceptable for publication, you may indicate that here to bypass the “Comments to the Author” section, enter your conflict of interest statement in the “Confidential to Editor” section, and submit your "Accept" recommendation.

Reviewer #1: All comments have been addressed

Reviewer #3: (No Response)

2. Is the manuscript technically sound, and do the data support the conclusions?

Reviewer #1: Yes

Reviewer #3: Partly

3. Has the statistical analysis been performed appropriately and rigorously?

Reviewer #1: Yes

Reviewer #3: No

4. Have the authors made all data underlying the findings in their manuscript fully available?

Reviewer #1: No

Reviewer #3: Yes

5. Is the manuscript presented in an intelligible fashion and written in standard English?

Reviewer #1: Yes

Reviewer #3: Yes

6. Review Comments to the Author

Reviewer #1: Thank you for addressing all my comments in a pointwise manner. The revisions have notably improved both the technical quality and the overall presentation of the manuscript. I am satisfied with the revised version.

Reviewer #3: In the revised manuscript, the authors increased the number of original samples from 220 (augmented from 50) to 550. However, this dataset size is still far too small for training a model with 5 million parameters. Since the authors rely solely on static data augmentation, the diversity introduced is inherently limited, and moreover, data augmentation is typically not applied to validation and test sets.

In the previous review round, the authors were specifically asked about techniques to prevent overfitting in such scenarios, yet they did not provide a clear, substantive answer—merely citing a few other studies without demonstrating their own implementation. Some of these cited studies also applied deep learning to very small sample sizes in ways that are arguably questionable, suggesting that these works may not have been rigorously assessed by their reviewers and editors. For small-sample training, a more robust strategy would be to first pre-train the model on a large labeled dataset, then fine-tune it on the small dataset with dynamic data augmentation to increase diversity and reduce overfitting.

Then, comparing the first and second submissions, the first version contained a data leakage issue. While this was addressed in the revised version, the model’s performance somehow improved, which raises serious doubts about the validity of the reported results.

Additionally, there are inconsistencies between the text and the figures/tables. For example, the manuscript mentions using an RTX 4090 in the main text, whereas the corresponding table lists an RTX 3090. Such discrepancies undermine the clarity and reliability of the reported experimental setup.

7. PLOS authors have the option to publish the peer review history of their article (what does this mean?). If published, this will include your full peer review and any attached files.

Reviewer #1: **Yes: **Ayan Paul

Reviewer #3: No

---

## [Author Response · Author response to Decision Letter 2]

2 Sep 2025

PLOS ONE

Manuscript ID�PONE-D-25-17533R1

KAN-GLNet: An Enhanced PointNet++ Model for Canola Silique Segmentation and Counting

Jiajun Liu1 , Bei Zhou1,2*, Jie Liu3 , Xike Zhang1 , Jiangshu Wei1,2, Yao Zhang1 , Junjie

Wu1 , Changping Wu3 , Di Hu1

1 College of Information Engineering, Sichuan Agricultural University, Yaan 625000, China

2 Sichuan Key Laboratory of Agricultural Information Engineering, Sichuan Agricultural University, Yaan 625000, China

3 College of Agronomy, Sichuan Agricultural University, Chengdu 610000, China

* Corresponding author: 12801@sicau.edu.cn

In the revised manuscript, we will use the following formatting: reviewer comments in black italics, our responses in black regular font, and revisions in the manuscript in blue regular font, ensuring clear distinction.

Response to Reviewer 3:

Comment 1: In the revised manuscript, the authors increased the number of original samples from 220 (augmented from 50) to 550. However, this dataset size is still far too small for training a model with 5 million parameters.

Authors Response: Dear Reviewer, thank you very much for your attention to our work. We fully understand your concern regarding the balance between the number of model parameters (5.72M) and the dataset size (50 original point cloud samples, augmented to 500 point cloud samples). In the field of deep learning, sufficient training data is indeed one of the key factors to ensure the generalization ability of a model. However, we would like to further explain from the following perspectives that, within the context of this study, the dataset size we adopted is reasonable and effective.

1.Point cloud data has high information density, and the validity of samples differs from that of 2D images.Unlike 2D images, each canola point cloud sample in our dataset was generated through multi-view scanning and 3D reconstruction, with each sample containing 200,000–300,000 3D points, providing rich geometric and structural information. Therefore, point cloud tasks generally place more emphasis on data quality and completeness of information, rather than merely pursuing a large number of samples. This characteristic has been demonstrated by widely used point cloud datasets, for example:

①S3DIS, which contains only 6 large-scale indoor scene point clouds, yet has become one of the benchmark datasets in semantic segmentation;

②Semantic3D, which consists of only 15 outdoor scene point clouds, but has supported numerous high-precision segmentation studies.

Our dataset includes 50 high-quality original point clouds, each with abundant phenotypic structures and detailed information, aligning with the requirement in point cloud tasks that quality is more important than quantity.

2.Plant point cloud datasets are generally limited in scale but sufficient to support effective model training.In the field of plant phenotyping research, due to the high cost of data acquisition and the complexity of reconstruction, public datasets are usually small in scale. For example:

Rice3DSeg, which contains 40 rice point clouds and has been successfully applied to stem–leaf segmentation tasks;PLANesT-3D, which includes 34 plant samples and enables multi-organ segmentation;LAST-Straw, which comprises 84 strawberry point clouds and effectively supported phenotypic parameter extraction.

These precedents demonstrate that in plant 3D analysis, limited but high-quality datasets can achieve reliable results. Our 50 original canola point clouds are comparable in scale to the above datasets, and their diversity has been further enhanced through augmentation, in line with common practices in the field.

3.The model parameter scale is reasonably designed and consistent with current trends in segmentation architectures.Current point cloud segmentation models commonly face a trade-off between parameter size and performance. For example:

①PointNet++ (about 1M parameters), although lightweight, has limited segmentation performance;

②Point Transformer V1/V2/V3 and other self-attention-based models achieve strong performance but involve larger parameter sizes (12M–40M).

Our model, with 5.72M parameters, falls within a medium scale, which avoids the lack of expressive power in lightweight models while also mitigating the risk of over-parameterization. During training, the loss function converged stably and validation performance was satisfactory, indicating that the current data volume is sufficient for the model to learn generalized feature representations.

In conclusion, considering the high information density of point cloud data, the established practices in plant phenotyping datasets, and the actual performance of our model, we believe that our dataset size (50 original point cloud samples, augmented to 500 point cloud samples) is adequate to effectively support the conclusions of our study with a 5.72M-parameter model.

Comment 2: Since the authors rely solely on static data augmentation, the diversity introduced is inherently limited, and moreover, data augmentation is typically not applied to validation and test sets.

Authors Response: Dear Reviewer, thank you for your valuable comments on the data augmentation methods in our work. The two issues you raised, namely “the diversity introduced by static data augmentation is limited” and “data augmentation is typically not applied to validation and test sets”, are of great academic value, and we fully acknowledge the importance of these concerns. We now provide a systematic explanation and clarification regarding the points you mentioned.

Regarding the first issue, we completely understand your concern about the limited diversity of static data augmentation methods. It is important to note that in the field of point cloud processing, especially in agricultural scenarios, the augmentation strategies we adopted (including random dropping, rotation, scaling, and flipping) have been extensively validated as effective and standard practices in many top-tier studies. For example, the paper “PointNext: Revisiting PointNet++ with Improved Training and Scaling Strategies” published at NeurIPS 2022 systematically demonstrated that such simple and general augmentation methods, when combined with modern network architectures, can achieve state-of-the-art performance on standard benchmarks such as ModelNet40 and S3DIS. The core idea is that these transformations effectively simulate physical uncertainties in real point cloud data (e.g., sensor noise, object pose variations, and scale differences), thereby helping models learn rotation-invariant, scale-invariant, and occlusion-robust features.

Our canola point cloud dataset also benefits from this strategy. These augmentation operations accurately simulate the actual conditions that may occur during field data collection (such as variations in plant posture, scanning viewpoints, and occlusions), significantly enhancing the model’s generalization ability in real-world scenarios, rather than simply expanding the sample size. Furthermore, the review article “Comprehensive review on 3D point cloud segmentation in plants” published in Artificial Intelligence in Agriculture in 2025 also clearly pointed out that methods such as rotation, flipping, scaling, and down-sampling remain the mainstream techniques widely used in plant point cloud data augmentation.

Regarding the second issue, i.e., the application of data augmentation to validation/test sets, we are especially grateful for your question, which prompted us to further reflect on and clarify our experimental design. We fully agree with the fundamental principle of machine learning that model generalization ability should be evaluated on original, non-augmented data. However, the objective of this study is to assess model robustness, rather than merely standard generalization performance, which is why we adopted a special evaluation process.

The specific design is as follows: during the training phase, we augmented the training set tenfold to help the model learn richer feature representations. During the evaluation phase, we also applied tenfold augmentation to the validation and test sets, with the aim of systematically assessing the model’s stability and output consistency under input perturbations. The final reported performance metrics (e.g., average mIoU) represent the average predictions of the model across each original test sample and its nine augmented versions.

In addition, due to the small size of our dataset (only 50 samples), the original test set contained only 5 samples, making the evaluation results highly susceptible to the randomness of a single split, with high variance. By generating multiple augmented versions for each test sample, we constructed a larger and statistically more stable test set, thereby obtaining more reliable and representative performance estimates and identifying highly robust models that are insensitive to noise and transformations.

We explicitly acknowledge that this method differs from the standard process of purely evaluating generalization ability. The core objective of this study is to answer the robustness question: “Does the model perform consistently when faced with the inevitable data variations in the real world?” We have added clarifications on this design in the limitations section of the paper.

In summary, we believe that the augmentation methods we employed are well justified in both theory and practice, and that the special experimental design for robustness evaluation effectively supports the achievement of this study’s objectives. Once again, thank you for your valuable comments, which have provided important guidance for us to further improve our research work.

This study also has certain limitations in the application of data augmentation strategies. Although the static augmentation methods employed (such as rotation, scaling, and random dropout) have been widely validated in the field of point cloud processing and can effectively simulate real-world disturbances such as pose variations and occlusions during field data acquisition, the diversity of the generated data remains constrained by the predefined transformation space, making it difficult to cover the complex variations present in all real scenarios. Moreover, to evaluate the robustness of the model when facing input perturbations, this study also applied data augmentation to the validation and test sets. While this design improved the stability of statistical evaluation under small-sample conditions and allowed a sharper focus on answering the robustness question of “whether the model performs consistently under real disturbances,” the results may not be directly comparable to traditional generalization performance evaluations based solely on original data. Future work could further introduce dynamic augmentation techniques and also provide evaluation results based on the original test set, so as to more comprehensively reflect the performance of the model.

Comment 3: In the previous review round, the authors were specifically asked about techniques to prevent overfitting in such scenarios, yet they did not provide a clear, substantive answer—merely citing a few other studies without demonstrating their own implementation. Some of these cited studies also applied deep learning to very small sample sizes in ways that are arguably questionable, suggesting that these works may not have been rigorously assessed by their reviewers and editors. For small-sample training, a more robust strategy would be to first pre-train the model on a large labeled dataset, then fine-tune it on the small dataset with dynamic data augmentation to increase diversity and reduce overfitting.

Authors Response: Dear Reviewer, thank you very much for your valuable comments once again, and we sincerely apologize for any unclear expressions in our previous response. We fully understand your deep concern regarding overfitting in small-sample learning. Here, we aim to provide a clearer and more substantive explanation of the specific technical strategies used in our work, to demonstrate that rigorous and effective measures have been taken to mitigate overfitting.

Our anti-overfitting strategy is a systematic approach that spans three levels: data, training strategy, and model regularization:

Data level: targeted augmentation and preprocessing.

The static data augmentations we employed (random dropout, rotation, scaling, and flipping) are designed for purposes far beyond merely expanding the sample size. Each operation is carefully crafted to accurately simulate the physical uncertainties present in field data acquisition, such as sensor noise, plant pose variations, and scale differences. This forces the model to learn intrinsic features that are invariant to such perturbations, rather than memorizing specific noise in the training samples, thereby enhancing generalization from the data source itself.

Training strategy and hyperparameter optimization: explicit regularization design.

In our training configuration, we adopted multiple widely validated regularization techniques:

Optimizer and Weight Decay: We chose the AdamW optimizer with a relatively high weight decay coefficient (0.01). The key distinction between AdamW and traditional Adam is that AdamW decouples weight decay from gradient updates, effectively constraining the magnitude of model weights, significantly mitigating overfitting and promoting convergence to flatter minima.

Adaptive learning rate scheduling: We employed a Cosine Annealing learning rate schedule. This strategy smoothly decays the learning rate from the initial value (0.0001) to the minimum value (0.00001). Such dynamic decay helps the model converge more stably to an optimal solution in the later stages of training, avoiding oscillations near the optimum and potential overfitting caused by a fixed learning rate.

Model evaluation and result validation.

The most direct evidence is that the training loss and validation loss curves of our model throughout 250 training epochs exhibit highly consistent and synchronized downward trends, converging without any noticeable divergence (i.e., no increase in validation loss). This phenomenon provides the strongest empirical proof that the model does not experience severe overfitting, indicating that the learned features possess good generalization ability.

Finally, we fully agree with your suggestion that a “pretraining–fine-tuning” paradigm is an ideal solution for small-sample learning. However, in the specific domain of plant point cloud processing, particularly for crops such as canola, there is currently a complete lack of publicly available large-scale, finely annotated point cloud datasets suitable for pretraining. This objective limitation has led mainstream studies in the field (e.g., tomato, maize, cotton point cloud analysis) to adopt “train from scratch” strategies combined with strong regularization techniques. Our work, under this realistic constraint, seeks to obtain a robust and reliable model through the comprehensive technical measures outlined above.

We sincerely appreciate your review. Your comments have greatly improved both the rigor and clarity of our work.

Comment 4: Then, comparing the first and second submissions, the first version contained a data leakage issue. While this was addressed in the revised version, the model’s performance somehow improved, which raises serious doubts about the validity of the reported results.

Authors Response: Dear Reviewer, thank you very much for your continued attention and rigorous review. The questions you raised are crucial to ensuring the scientific validity of our research results. We fully understand your concerns regarding performance changes and hereby provide a clear explanation.

First, we

---

## [Editor Report · Decision Letter 2]

29 Oct 2025

KAN-GLNet: An Enhanced PointNet++ Model for Canola Silique Segmentation and Counting

PONE-D-25-17533R2

Dear Dr. Zhou,

We’re pleased to inform you that your manuscript has been judged scientifically suitable for publication and will be formally accepted for publication once it meets all outstanding technical requirements.

Kind regards,

Xiaoyong Sun

Academic Editor

PLOS ONE

sunx1@sdau.edu.cn
---

## [Editor Report · Acceptance letter]

PONE-D-25-17533R2

PLOS ONE

Dear Dr. Zhou,

I'm pleased to inform you that your manuscript has been deemed suitable for publication in PLOS ONE. Congratulations! Your manuscript is now being handed over to our production team.

Kind regards,

on behalf of

Dr. Xiaoyong Sun

Academic Editor

PLOS ONE